EMBO
Molecular Medicine

# Antiviral drugs prolong survival in murine recessive dystrophic epidermolysis bullosa

Grace Tartaglia [1], Ignacia Fuentes [2,3,4], Neil Patel[5], Abigail Varughese[1], Lauren E Israel [1],
Pyung Hun Park[1], Michael H Alexander [1], Shiv Poojan[1], Qingqing Cao [1], Brenda Solomon[1],
Zachary M Padron[1], Jonathan A Dyer[6], Jemima E Mellerio[7], John A McGrath[7], Francis Palisson [2,8],
Julio Salas-Alanis[9], Lin Han[5] & Andrew P South [1,10,11,12] ✉

## Abstract

**Recessive dystrophic epidermolysis bullosa (RDEB) is a rare inherited skin disease characterized by defects in type VII collagen leading to a range of fibrotic pathologies resulting from skin fragility, aberrant wound healing, and altered dermal fibroblast physiology. Using a novel in vitro model of fibrosis based on endogenously produced extracellular matrix, we screened an FDA-approved compound library and identified antivirals as a class of drug not previously associated with anti-fibrotic action. Preclinical validation of our lead hit, daclatasvir, in a mouse model of RDEB demonstrated significant improvement in fibrosis as well as overall quality of life with increased survival, weight gain and activity, and a decrease in pruritus-induced hair loss. Immunohistochemical assessment of daclatasvir-treated RDEB mouse skin showed a reduction in fibrotic markers, which was supported by in vitro data demonstrating TGFβ pathway targeting and a reduction of total collagen retained in the extracellular matrix. Our data support the clinical development of antivirals for the treatment of patients with RDEB and potentially other fibrotic diseases.**

**Keywords** Fibrosis; Collagen; Recessive Dystrophic Epidermolysis Bullosa; Drug Repurposing; Antivirals
**Subject Categories** Pharmacology & Drug Discovery; Skin

## Introduction

Around half a million people worldwide suffer from epidermolysis bullosa (EB), a rare group of genetic diseases characterized by fragile skin with the involvement of other epithelial tissues (Tabor et al, 2017). Of the four main categories of EB, recessive dystrophic EB, or RDEB, is burdened by extensive scarring and fibrosis. RDEB is also complicated by lethal cutaneous squamous cell carcinoma (cSCC) and is caused by pathogenic variants in the gene *COL7A1*. *COL7A1* encodes type VII collagen, a constituent component of anchoring fibrils which are supermolecular structures that strengthen adhesion between a stratifying epithelium and the underlying stroma. Anchoring fibrils are particularly prevalent in the skin and absence or incomplete formation of anchoring fibrils leads to fragility and results in painful blisters and hard to heal wounds upon minimal mechanical trauma (Pfendner and Lucky, 2018). In addition to structural support, type VII collagen also supports normal fibroblast physiology and its absence leads to major signaling changes culminating in increased transforming growth factor beta, or TGFβ. The constant wounding, healing, and inflammation that RDEB patients experience coupled with altered fibroblast physiology, results in substantial fibrosis which has been identified as a major factor driving the development of lethal cSCC. Median survival of severe RDEB patients after diagnosis of their first SCC is 2.4 years (Robertson et al, 2021), which is in stark contrast to cSCC arising in the general population.

Fibrosis is characterized by excessive extracellular matrix (ECM) deposition and disorganization of collagen fibers which generates stiffer and less resilient tissue structure that is often progressive due to feed forward of major fibrosis driving signaling pathways. One of the significant drivers of fibrosis is elevated TGFβ signaling and stiffer tissues lead to increased TGFβ due in part to greater accessibility to the ligand (Biernacka et al, 2011; Wells and Discher, 2008). TGFβ is a multifunctional cytokine that is important in regulating crucial cellular activities, and has canonical and non-canonical pathways dictated by SMAD dependency. The SMADs are a group of intracellular proteins that transmit signals from TGFβ at the cell surface onto the nucleus. In the SMAD-dependent pathway, SMADs are phosphorylated by the kinase domain of the TGFβ receptor tetramer and can interact with transcription

[1]Department of Pharmacology, Physiology, and Cancer Biology, Thomas Jefferson University, Philadelphia, PA, USA. [2]DEBRA Chile, Santiago, Chile. [3]Departamento de Biología Celular y Molecular, Facultad de Ciencias Biológicas, Pontificia Universidad Católica de Chile, Santiago, Chile. [4]Centro de Genética y Genómica, Facultad de Medicina Clínica Alemana, Universidad de Desarrollo, Santiago, Chile. [5]School of Biomedical Engineering, Science and Health Systems, Drexel University, Philadelphia, PA, USA. [6]Department of Dermatology, University of Missouri School of Medicine, Columbia, MO, USA. [7]St. John's Institute of Dermatology, King's College London (Guy's Campus), London, UK. [8]Servicio de Dermatologia, Facultad de Medicina Clínica Alemana-Universidad de Desarrollo, Santiago, Chile. [9]Instituto Dermtaologico de Jalisco, Guadalajara, Mexico. [10]The Joan and Joel Rosenbloom Research Center for Fibrotic Diseases, Thomas Jefferson University, Philadelphia, PA, USA. [11]Sidney Kimmel Cancer Center, Thomas Jefferson University, Philadelphia, PA, USA. [12]Department of Otolaryngology Head and Neck Surgery, Thomas Jefferson University, Philadelphia, PA, USA. ✉E-mail: Andrew.south@Jefferson.edu

cofactors to modulate target genes, including many ECM components such as collagen I (Attisano and Lee-Hoeflich, 2001). SMAD3 has a role in mediating cutaneous wound healing, as seen in a SMAD3-null mouse model that experiences accelerated healing compared to wild-type mice (Ashcroft and Roberts, 2000; Ashcroft et al, 1999; Hu et al, 2003; Walton et al, 2017). The SMAD-independent pathway, on the other hand, includes alternative non-canonical pathways such as the PI3K/AKT pathway, and many others that contribute to a complex network of regulation (Clayton et al, 2020; Trojanowska, 2009; Zhang, 2009).

Drug repurposing, a process where previously approved and regulated compounds are tested for potential efficacy in new indications, represents a cost-effective and, more importantly, rapid approach to developing therapies for disease (Parvathaneni et al, 2019; Pushpakom et al, 2019; Roessler et al, 2021). Here, we take such an approach by using a novel assay of fibrosis based on patient primary cells in culture to screen a library of 1443 FDA-approved compounds and identify 43 hits with potential efficacy in RDEB. Among these hits were multiple antiviral drugs, which as a category of compounds were previously unidentified as having anti-fibrotic efficacy. We go on to show our lead-hit antiviral compound, daclatasvir, improved multiple quality-of-life metrics in a mouse model of RDEB paving the way for clinical development of this and related drugs for patient treatment.

## Results

### ECM produced by primary RDEB dermal fibroblasts detaches from tissue culture plastic faster than non-RDEB ECM

We previously demonstrated that RDEB dermal fibroblasts in culture produce an altered ECM compared with non-RDEB dermal fibroblasts (Ng et al, 2012) and that RDEB ECM generates altered collagen fibril density after remodeling in suspension culture over time (Atanasova et al, 2019). During this work, we noticed that RDEB matrices would often spontaneously detach from tissue culture plastic much earlier than non-RDEB matrices (Appendix Fig. S1) and that the readout of time to detachment for fibroblast ECM could be utilized for a medium-throughput screen to identify novel compounds that altered ECM production. To begin with we compared primary dermal fibroblast matrices from RDEB and non-RDEB patients and observed significantly shorter times to detach in RDEB (Fig. 1A), 10 days on average compared to 17 days, respectively. Since RDEB fibroblasts have more endogenous TGFβ signaling than non-RDEB fibroblasts (Akasaka et al, 2021; Atanasova et al, 2019) and TGFβ is a known driver of fibrosis, we investigated matrix detachment in response to TGFβ stimulation and inhibition. SB431542, a TGFβ1 receptor inhibitor, and SIS3, a SMAD3 inhibitor, delayed detachment in RDEB matrices (Fig. 1B,C) while exogenous TGFβ1 accelerated detachment in non-RDEB matrices (Fig. 1D). Next, we explored the mechanism of action for accelerated detachment in RDEB by first comparing cell viability to test the hypothesis that changes in cell number influence the amount of ECM present in the culture dish. In agreement with previous studies (Ng et al, 2012), we observed similar cell viability over a 10-day span, the time it takes for RDEB matrices to detach (Fig. 1E).

Since tension could influence ECM detachment from tissue plastic, we used atomic force microscopy to measure the elastic modulus (a measure of stiffness under strain) of RDEB and non-

RDEB ECM. We observed a higher elastic modulus, indicative of increased stiffness, in RDEB matrices and that exogenous TGFβ treatment increased elastic modulus (Fig. 1F,G). With this knowledge, we measured the hydroxyproline content of matrices, since collagen contributes the majority of hydroxyproline in the ECM and collagen content and density are known to influence elastic modulus (Calò et al, 2020). We compared hydroxyproline levels in the matrix (representing insoluble proteins, the majority of which are assumed to be processed collagen present in the ECM) with the amount of hydroxyproline present in the cell media (representing soluble, unprocessed collagen, but also measuring other ECM proteins such as glycoproteins (Küttner et al, 2013)). To measure hydroxyproline we used a colorimetric assay and to control for whether the hydroxyproline content adhered to the tissue culture plastic is intracellular or matrix in origin we first removed the cells using an ammonia wash (Akasaka et al, 2021) after 7 days and compared the two conditions, determining that the majority of hydroxyproline content is in the matrix (Fig. 1H). We then measured hydroxyproline in 7-day matrix cultures and noted an increase in hydroxyproline in RDEB matrices compared to non-EB matrices, which is in line with our observations that RDEB matrices form and detach faster than non-EB matrices. We also measured the level of hydroxyproline in the media at day 7 and noted a greater proportion of soluble collagen in the media of non-EB cultures presumably as a result of reduced matrix incorporation. Interestingly, TGFβ inhibition with the ALK5 inhibitor SB431542 reversed the ratio of matrix to media hydroxyproline in RDEB, as did adding exogenous TGFβ1 to non-EB matrices (Fig. 1I). These data suggest that TGFβ signaling and collagen content in the ECM influences both ECM tensile strength and time to detachment of primary dermal fibroblast-derived matrices.

### Collagen I levels correlate with fibroblast matrix detachment

Next, we measured detachment and levels of collagen I, the major collagen present in skin, as well as TGFβ1 in an isogenic setting to control for potential inter-patient variability. We examined sample sets of biopsy-derived fibroblasts from SCC tumor and non-tumor regions of three individual RDEB patients, focusing initially on patient RDEB119 from whom multiple biopsy sites were available including distal "normal" skin (Fig. 2A). Tumor-derived fibroblasts, known to have the greatest alterations in the expression of ECM components in RDEB (Ng and South, 2014), detached faster than fibroblasts from peri-tumor or sites distal to the tumor. Tumor-derived fibroblasts had a greater amount of endogenous TGFβ1 mRNA and collagen I mRNA and protein, compared with distal from patient RDEB119 (Fig. 2B–D). The data for collagen I were confirmed in isogenic sets from patients 2 and 3 with the exception that no distal sites were available (Fig. 2E,F).

### FDA-approved compound library screen identifies antiviral drugs as a novel class of anti-fibrotic compounds

Using matrix detachment as a readout of fibrosis in primary RDEB dermal fibroblasts, we screened 1443 FDA-approved compounds at 10 µM concentration over a 30-day period in RDEB fibroblast populations. We focused on those compounds that delayed

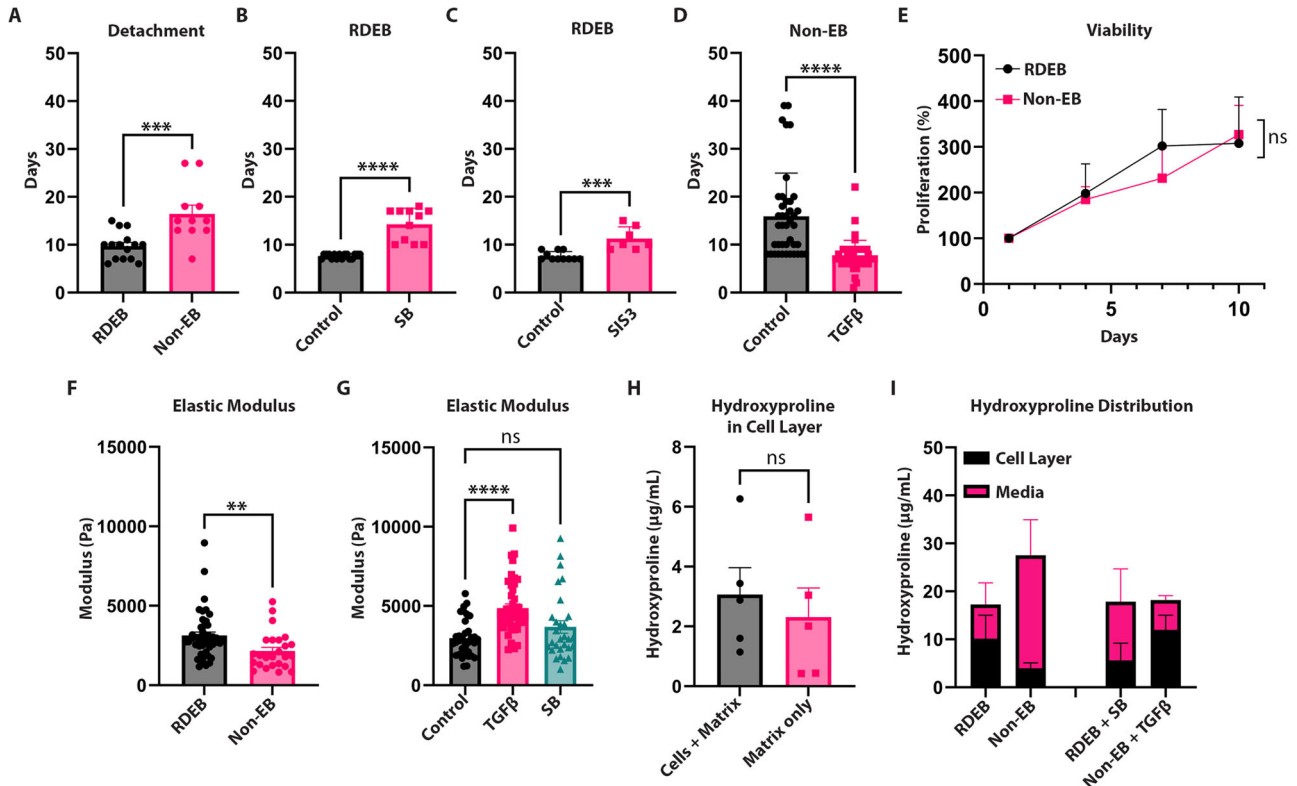

**Figure 1. Matrix detachment recapitulates fibrotic phenotypes through the TGFβ pathway.**

(A) Graph of time to matrix detachment in RDEB (n = 15, 7 biological replicates in 2–3 technical replicates) and non-EB (n = 11, 8 biological replicates in 1–2 technical replicates). Mann–Whitney test for significance. (B) Graph of time to matrix detachment in RDEB with vehicle control (n = 16, 5 biological replicates in 3–4 technical replicates) and SB431542 (n = 11, 5 biological replicates in 2–3 technical replicates). Mann–Whitney for significance. (C) Graph of time to matrix detachment in RDEB with vehicle control (n = 11, 5 biological replicates in 2–3 technical replicates) and SIS3 (n = 7, 5 biological replicates in 1–2 technical replicates). Mann–Whitney test for significance. (D) Graph of time to matrix detachment in non-EB with vehicle control (n = 41, 7 biological replicates in 6–7 technical replicates) and exogenous TGFβ1 (n = 42, 7 biological replicates in 6–7 technical replicates). Mann–Whitney test for significance. (E) Graph of RDEB (n = 4 biological replicates) and non-EB (n = 3 biological replicates) population viability over 10 days. Paired t test for significance. (F) Graph for elastic modulus of matrix in RDEB (n = 25, 7 biological replicates in 3–4 technical replicates) and non-EB (n = 45, 6 biological replicates in 7–8 technical replicates) matrix. Unpaired t test for significance. (G) Graph for elastic modulus of matrix in RDEB with vehicle control, exogenous TGFβ, and SB431542 (3 biological replicates in 12 technical replicates) treatment. Kruskal–Wallis test with Dunn's correction for significance. (H) Graph of total hydroxyproline amount in cell layer comparing cells and matrix (n = 5 biological replicates) and matrix only (n = 5 biological replicates). Unpaired t test for significance. (I) Graph of total hydroxyproline amount (mean ± SEM) in cell layer and media of RDEB (n = 6, 3 biological replicates in duplicate), non-EB (n = 6, 3 biological replicates in duplicate), RDEB + SB431542 (n = 6, 3 biological replicates in duplicate), and non-EB + TGFβ1 (n = 6, 3 biological replicates in duplicate) treatment. Data information: In (A–H), data are presented as mean ± SEM. **P ≤ 0.01, ***P ≤ 0.001, ****P ≤ 0.0001, P > 0.05 is not significant (ns). Source data are available online for this figure.

detachment but did not inhibit detachment completely since we wanted to identify compounds that normalize the detachment defect in RDEB in a similar manner to non-EB fibroblasts, without inhibiting matrix production or target cell viability. We used 10 µM for the initial screen with the intent to avoid identifying compounds with potential toxicity that might exacerbate an already devastating phenotype in patients with RDEB. The primary detachment screen in one RDEB population identified 101 compounds that delayed detachment, while 78 compounds accelerated detachment. We also noted that 1040 compounds detached at the same time as the vehicle control, 118 compounds killed the fibroblasts, and 106 compounds affected the matrix in variant "other" ways, such as abrogating matrix formation partially or entirely (Fig. 3A). From here, we took the 101 compounds that delayed detachment to a secondary screen with a different RDEB

patient population. Of these, we confirmed 43 compounds delayed detachment in both populations, 33 compounds detached at the same time as vehicle control, 6 compounds killed the fibroblasts, and 19 compounds had "other" matrix effects as previously described (Fig. 3B). The 43 compounds that delayed detachment in two RDEB populations (Appendix Table S1) had three major categories that dominated the hit drugs: steroids, kinase inhibitors, and antivirals/antimalarials (Fig. 3C). The steroid and kinase inhibitor hits provided validation as positive controls since all have published literature supporting their anti-fibrotic effects (Mendoza and Jimenez, 2022; Oku et al, 2008), further validating our assay as a fibrosis surrogate. Among the steroid hits, halcinonide consistently and significantly delayed detachment compared to vehicle control in RDEB fibroblasts (Fig. 3D). Another steroid hit, fluticasone propionate, delayed detachment but also significantly

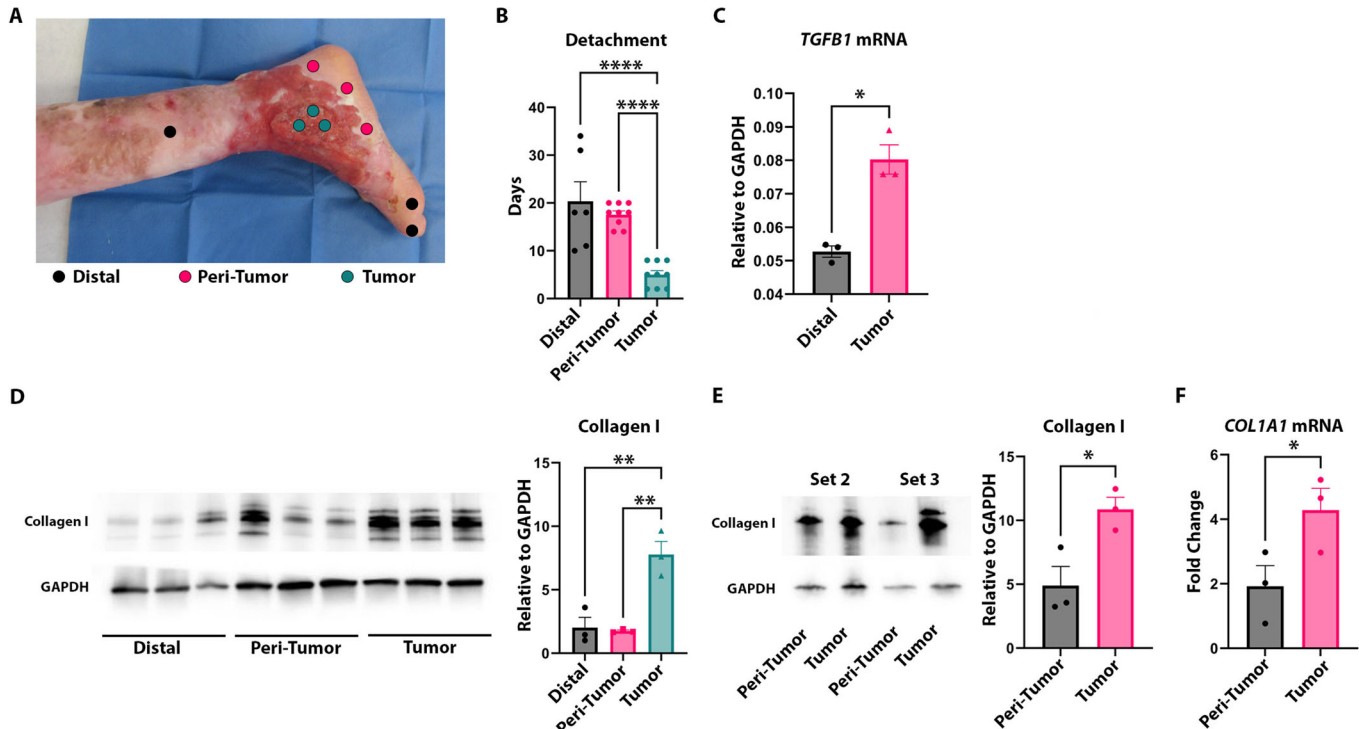

**Figure 2. Isogenic sample sets identify collagen I a potential driver of matrix detachment.**

(A) Visual of amputated leg of RDEB patient. Biopsy sites are shown as circles, from tumor, peri-tumor, and distal sites. (B) Graph of time to matrix detachment in RDEB fibroblasts from tumor ($n = 9$), peri-tumor ($n = 9$), and distal ($n = 6$) biopsy sites. Ordinary one-way ANOVA with Holm–Šídák test for significance. (C) Graph of log2 values of TGFβ1 mRNA from fibroblasts in distal ($n = 3$) and tumor ($n = 3$) biopsies. Welch's $t$ test for significance. (D) Western blot of collagen I and GAPDH in distal ($n = 3$), peri-tumor ($n = 3$), and tumor ($n = 3$) fibroblasts (left) and quantification of blot presented as graph showing collagen I expression relative to GAPDH. Ordinary one-way ANOVA with Šídák test for significance. (E) Western blot of collagen I and GAPDH in peri-tumor ($n = 3$) and tumor ($n = 3$) fibroblasts in isogenic patient sample sets 2 and 3 (left) and quantification of blot presented as graph of collagen I expression relative to GAPDH (right). Welch's $t$ test for significance. (F) qPCR result of COL1A1 mRNA presented as graph of fold change relative to GAPDH mRNA in peri-tumor ($n = 3$ biological replicates) and tumor ($n = 3$ biological replicates). Paired $t$ test for significance. Data information: In (B–F), data are presented as mean ± SEM. *$P \le 0.05$, **$P \le 0.01$, ****$P \le 0.0001$. Source data are available online for this figure.

inhibited cell viability after 7 days in culture (Fig. 3E). Ripasudil significantly delayed detachment compared to control (Fig. 3F), while idelalisib significantly inhibited fibroblast viability (Fig. 3G). Daclatasvir, idoxuridine, and sofosbuvir consistently delayed detachment in both RDEB patient populations, while chloroquine, mefloquine, and rilpivirine had a wide range in response between the populations (Fig. 3H). Interestingly, none of the antivirals or antimalarials significantly inhibited fibroblast viability even at a higher concentration of 10 μM (Fig. 3I). Of the "other" drugs that could not be commonly categorized together, most exhibited a wide range in response between the RDEB patient populations (Fig. 3J). Of these drugs, deferasirox and eltrombopag olamine significantly inhibited fibroblast viability (Fig. 3K). Overall, the hit compounds we describe here are delaying detachment with the majority not affecting the cell proliferation capability, implicating a matrix-specific mechanism of action.

To confirm antiviral compounds and to identify further drugs in this class we performed a smaller screen of a focused 240 antiviral drug library and found that there were 57 hits across two RDEB populations that delayed matrix detachment (Appendix Fig. S2A) and narrowed down our list of hits to thirty-two compounds (Appendix Table S2) that potentially are targeting the fibrotic ECM of RDEB fibroblasts without affecting cell proliferation (Appendix Fig. S2B).

## Antivirals downregulate TGFβ signaling and collagen retention in the matrix

The six antiviral compounds identified from the initial screen were: rilpivirine, sofosbuvir, mefloquine, chloroquine, idoxuridine, and daclatasvir. We identified daclatasvir as a lead hit since dose–response experiments showed this drug to be most effective at a lower concentration (1 μM) than what was used for the screen (10 μM) and it was also the most effective drug without inducing accelerated detachment at a higher concentration (Appendix Fig. S3). We also pursued idoxuridine in subsequent experiments since this drug showed the most similar detachment trends to Daclatasvir in the dose–response detachment experiments. Across multiple RDEB populations, idoxuridine and daclatasvir delayed detachment at 1 μM compared to untreated control (Fig. 4A), but this was not evident in non-RDEB populations (Appendix Fig. S4A), indicating that the ability of daclatasvir to delay detachment was RDEB-specific in these experiments. We also observed no significant change in RDEB cell viability (Fig. 4B) or in non-RDEB cell viability (Appendix Fig. S4B) over 10 days compared to untreated control. Using Western blotting of total cell lysates, we observed that idoxuridine and daclatasvir inhibited the TGFβ pathway targets collagen I, phosphorylated AKT, and phosphorylated SMAD3 in EB but not in non-EB (Fig. 4C; Appendix Fig. S4C) at the proteomic level and COL1A1

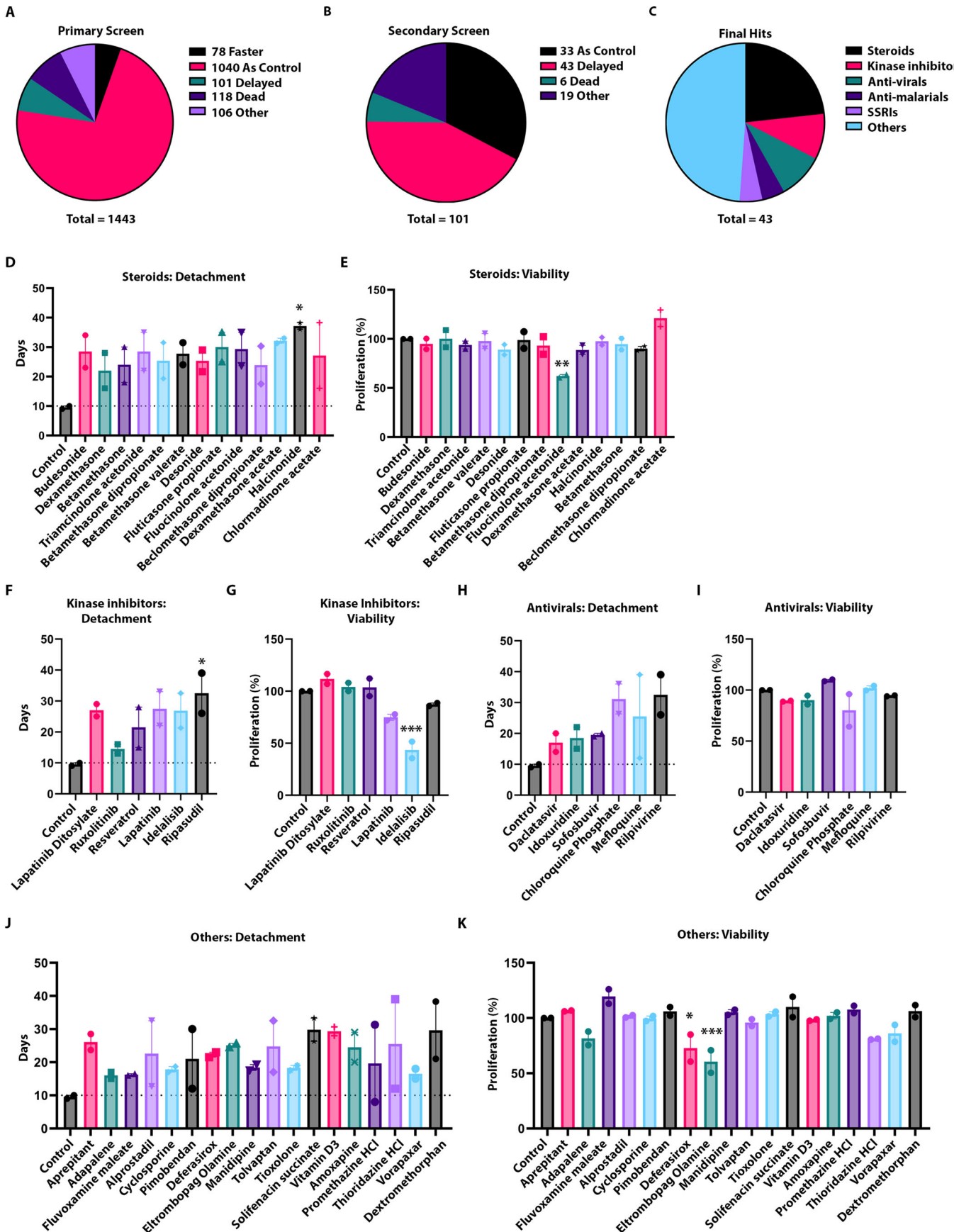

◄ **Figure 3. Antivirals are a novel class of anti-fibrotic compounds in RDEB.**

(A) Pie chart visual showing detachment trends across the original 1443 compounds in one RDEB fibroblast matrix assay. (B) Pie chart visual showing detachment trends in 170 compounds in a second RDEB fibroblast matrix assay. (C) Pie chart visual showing drug class hit-targets from both detachment assay screens. (D) Graph of detachment assay results of steroid hits that delayed detachment in both RDEB patient populations ($n = 2$, 2 biological replicates in 2–3 technical replicates). (E) Graphs of viability from steroid hits after 7 days ($n = 2$, 2 biological replicates in 2–3 technical replicates). (F) Graph of detachment assay results of kinase inhibitor hits that delayed detachment in both RDEB patient populations ($n = 2$, 2 biological replicates in 2–3 technical replicates). (G) Graphs of viability from kinase inhibitor hits after 7 days ($n = 2$, 2 biological replicates in 2–3 technical replicates). (H) Graph of detachment assay results (mean ± SEM) of antiviral and antimalarial hits that delayed detachment in both RDEB patient populations ($n = 2$, 2 biological replicates in 2–3 technical replicates). (I) Graphs of viability (mean ± SEM) from antiviral and antimalarial hits after 7 days ($n = 2$, 2 biological replicates in 2–3 technical replicates). (J) Graph of detachment assay results (mean ± SEM) of non-grouped "other" hits that delayed detachment in both RDEB patient populations ($n = 2$, 2 biological replicates in 2–3 technical replicates). (K) Graphs of viability from non-grouped "other" hits after 7 days ($n = 2$, 2 biological replicates in 2–3 technical replicates). Data information: In (D–G, K), data are presented as mean ± SEM, and statistical analysis was performed with ordinary one-way ANOVA corrected with Dunnett's test. *$P \leq 0.05$, ***$P \leq 0.001$. Source data are available online for this figure.

at the mRNA level (Fig. 4D). When we measured hydroxyproline levels in the media compared to the matrix, we found that daclatasvir promoted the presence of hydroxyproline in the media in RDEB and reduced the levels in the matrix similar to the proportion seen in non-RDEB; however, while idoxuridine also mimicked the same increase of hydroxyproline in media in proportion to matrix in RDEB, it also caused collagen retention in the matrix in non-RDEB (Fig. 4E). Interestingly, while there is an observed, nonsignificant difference in the collagen retained in the RDEB matrix with both daclatasvir and idoxuridine, there is no change in the modulus of the matrix with treatment (Fig. 4F).

## Daclatasvir improves survival, increases weight, increases activity and reduces the development of fibrosis in RDEB mice

In order to evaluate the preclinical potential of daclatasvir for the prevention of fibrosis in patients with RDEB, we administered daclatasvir dihydrochloride in the drinking water of the hypomorphic mouse model of RDEB, which are also known as the *Col7a1*$^{flNeo/flNeo}$ mice, at a dose of 30 mg/kg. We found that daclatasvir greatly improved survival, with untreated control RDEB mice having a 4.9-fold increased chance of dying in their first 100 days of life compared to daclatasvir-treated mice (Fig. 5A). RDEB mice treated with daclatasvir not only lived longer, but also displayed more energetic activity at 60 days old (Fig. 5B), a strong indication of increased life quality. Another sign of improved life quality was increased weight retention, in which daclatasvir-treated mice on average weighed 31.4% more than control mice at death (Fig. 5C). We also noticed that the mice retained more of their hair with treatment, indicating less pruritus in the skin (and subsequent scratching) of the daclatasvir-treated group compared to control RDEB mice (Fig. 5D). Another noted phenomenon of the RDEB mice is their fusion of digits, which was delayed but not completely inhibited with daclatasvir treatment (Fig. 5E), potentially due to the increased activity of the mice. When we examined the mice for markers of fibrosis, we found that phospho-SMAD3 and collagen I expression were both reduced in the skin with daclatasvir treatment (Fig. 5F,G). Overall, RDEB mice treated with daclatasvir experienced increased life expectancy and quality of life as determined by weight gain, reduced hair loss and overall activity.

## Discussion

Fibrosis is a major, progressive disease that afflicts the general population even in first-world countries, and very few FDA-approved therapies are available to treat a pathology that manifests

as many different organ-specific diseases, such as pulmonary and hepatic fibrosis (Amati et al, 2023; Wollin et al, 2014). Only two drugs, pirfenidone and nintedanib, have been approved for the treatment of idiopathic pulmonary fibrosis, and no drugs have been approved for other fibrotic diseases. Arguably this is due to the lack of laboratory-based models and the need for organ specificity to support preclinical testing. Here we have developed an in vitro surrogate assay of fibrosis that differentiates disease pathology in primary dermal fibroblasts from patients with RDEB. We have used this model to screen FDA-approved compounds and have identified antivirals as a novel category of anti-fibrotic drugs with demonstrable efficacy in a preclinical mouse model (Fig. 6). While the use of antivirals has previously demonstrated anti-fibrotic effects in hepatitis C virus (HCV) infections (Calvaruso and Craxì, 2014; Cheng et al, 2021) these observations were attributed to the targeting of the underlying viral infection, and our work suggests a potentially novel mechanism for antiviral drugs through prevention of the fibrotic reaction by targeting TGFβ signaling.

Our hit compound of interest, daclatasvir, is an oral therapy previously used in combination to target HCV in hepatic fibrosis (Zakaria and El-Sisi, 2020) and while very few studies have investigated daclatasvir as a monotherapy, the drug shows very few interactions with commonly prescribed medications outside of metabolizing liver enzymes (Garimella et al, 2016). Daclatasvir has demonstrated rigorous safety standards and has shown good potency with a once-a-day oral dose (Gandhi et al, 2018; Gentile et al, 2014; Jafri and Gordon, 2015). Animal studies have shown that daclatasvir can cross the placenta during pregnancy and has also been found in small doses in lactation milk (Daclatasvir, 2015). Daclatasvir is 99% bound to maternal plasma proteins in breastmilk, but extensive studies have not been performed in nursing mothers undergoing HCV treatment (LactMed, 2006). Our approach to first treat pregnant dams with oral daclatasvir improved RDEB mouse longevity, life quality and slowed the onset of fibrosis, all of which provide preclinical evidence that this drug may have beneficial results for patient therapy.

We reveal that targeting TGFβ signaling is a novel mechanism of action for daclatasvir which was originally described as a potent inhibitor of the non-structural HCV protein 5A (NS5A) validated by experiments using specific NS5A resistant mutants and pull-down assays (Boson et al, 2017; Gao et al, 2010; O'Boyle Ii et al, 2013). Later experiments suggest that daclatasvir interferes with formation of double-membrane vesicles in the endoplasmic reticulum (ER) that contain the HCV RNA replication complex, and other work shows that daclatasvir decreases mobility and intracellular redistribution and/or clustering of NS5A (Ascher et al, 2014; Gao et al, 2010;

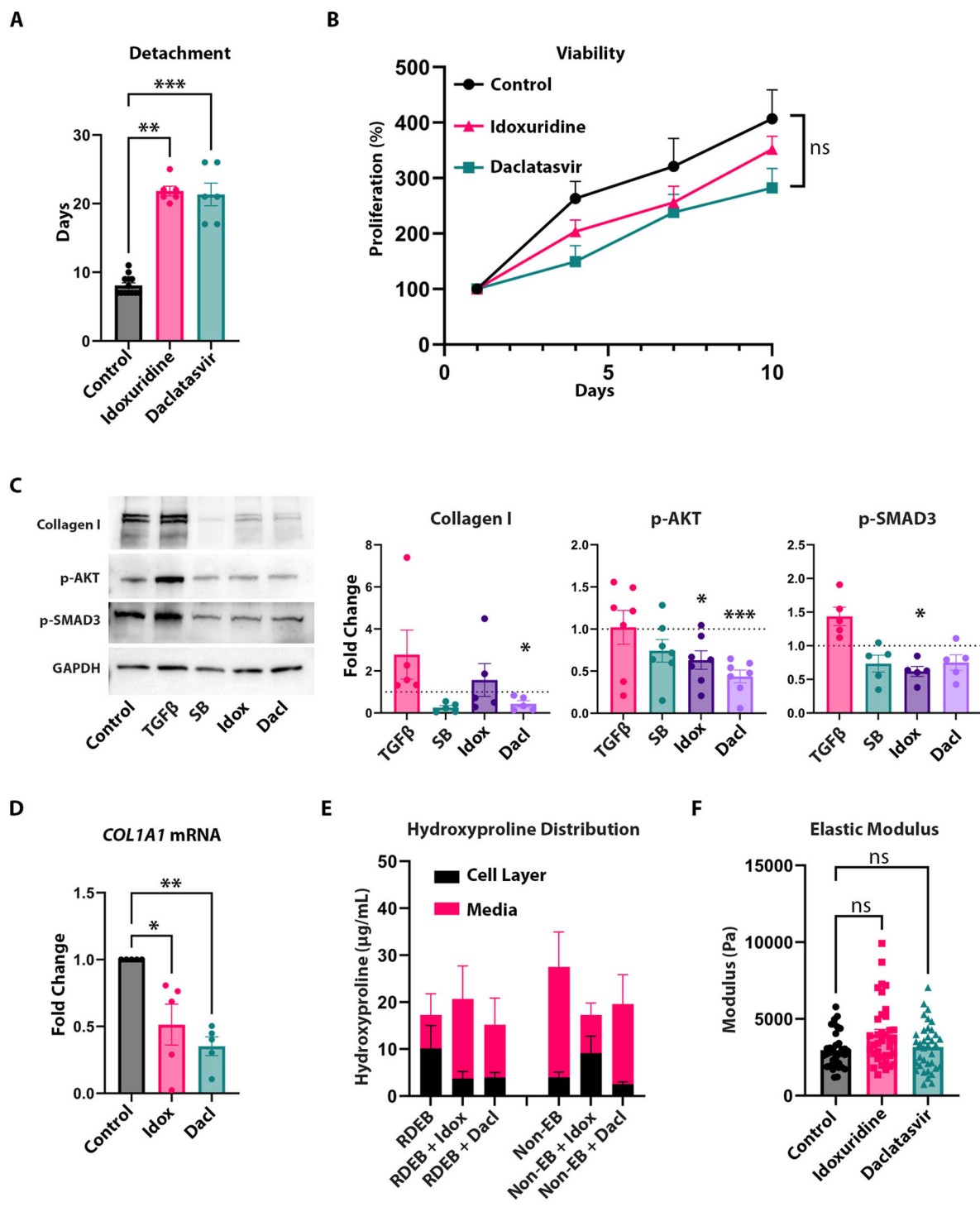

Gentile et al, 2014; He et al, 2006; O'Boyle Ii et al, 2013; Smith et al, 2016; Tellinghuisen et al, 2008), all of which point to perturbation of the ER membrane in the context of HCV. While it is tempting to speculate that daclatasvir may alter ER-Golgi traffic in RDEB, recently highlighted as a source of endogenous TGFβ signaling in RDEB fibroblasts (Cao et al, 2022), further work is needed to determine whether this is a mechanism of action for daclatasvir in RDEB.

Another potential mechanism of action for daclatasvir relates to NS5A being highly phosphorylated on serine/threonine residues (He et al, 2006; Tellinghuisen et al, 2008), similar to SMAD proteins, and daclatasvir targets the domain 1 region of NS5A that is imperative for viral RNA replication (Ascher et al, 2014). It is therefore possible to speculate that daclatasvir has an inhibitory role on phosphorylation, which would impact phosphorylation of the downstream TGFβ

◀ **Figure 4. Antivirals are a novel class of compounds with anti-fibrotic effects in RDEB.**

(A) Graph of time to matrix detachment in RDEB fibroblast populations comparing vehicle control ($n = 13$, 5 biological replicates in 2–3 technical replicates) to 1 μM idoxuridine ($n = 6$, 5 biological replicates in 1–2 technical replicates) and daclatasvir ($n = 6$, 5 biological replicates in 1–2 technical replicates) treatment. Kruskal–Wallis test with Dunn's correction performed for significance. (B) Line graph of RDEB cell proliferation (mean ± SEM) as a percentage compared to Day 1 with vehicle control ($n = 10$, 5 biological replicates in duplicate), idoxuridine ($n = 10$, 5 biological replicates in duplicate), and daclatasvir ($n = 10$, 5 biological replicates in duplicate). Ordinary one-way ANOVA with Dunnett test performed. (C) Western blots of collagen I, p-AKT, p-SMAD3, and GAPDH in RDEB fibroblasts with vehicle control, exogenous TGFβ, SB431542, idoxuridine, and daclatasvir treatment (left) and quantification of blots presented as graphs of collagen I ($n = 5$ biological replicates), p-AKT ($n = 7$ biological replicates), and p-SMAD3 ($n = 5$ biological replicates) relative to GAPDH. RM one-way ANOVA with Holm–Šídák test for p-AKT and p-SMAD3 and Dunnett test for collagen I. (D) qPCR result of COL1A1 mRNA presented as graph of Fold Change relative to GAPDH mRNA in RDEB fibroblasts ($n = 5$ biological replicates) with vehicle control, idoxuridine, and daclatasvir. RM one-way ANOVA with Holm–Šídák test for significance. (E) Graph of total hydroxyproline amount (mean ± SEM) in cell layer and media of RDEB ($n = 6$ biological replicates), RDEB + idoxuridine ($n = 6$ biological replicates), RDEB + Daclatasvir ($n = 6$ biological replicates), non-EB ($n = 6$ biological replicates), EB + idoxuridine ($n = 6$ biological replicates), and non-EB + Daclatasvir ($n = 6$ biological replicates) treatment. (F) Graph of elastic modulus of RDEB matrix with vehicle control ($n = 36$, 3 biological replicates in 12 technical replicates), idoxuridine ($n = 37$, 3 biological replicates in 12 technical replicates), and daclatasvir ($n = 36$, 3 biological replicates in 12 technical replicates) treatment. Kruskal–Wallis test with Dunn's correction performed for significance. Data information: In (A, C, D, F), data are presented as mean ± SEM. *$P ≤ 0.05$, **$P ≤ 0.01$, ***$P ≤ 0.001$, $P > 0.05$ is not significant (ns). Source data are available online for this figure.

targets, SMAD3 and AKT directly, and collagen I expression indirectly. While these observations (inhibition of phosphorylated SMAD3 and AKT) are consistent with idoxuridine, the other antiviral compound taken forward in vitro, further work needs to be performed to determine whether targeting TGFβ signaling is consistent for all antivirals as a class of fibrotic preventatives.

With regards to the mechanics of delayed matrix detachment with antiviral treatment of RDEB primary fibroblasts in vitro, we investigated multiple enzymes involved in the collagen processing pathway that could affect the ratio of soluble to insoluble collagen using both a targeted qPCR approach as well as RNA-sequencing (Appendix Fig. S5A,B). Lysyl oxidase (LOX), the enzymes responsible for crosslinking collagen, has been identified as upregulated in tumor-primed RDEB mice due to increased TGFβ signaling, and leading to increased ECM stiffness (Mittapalli et al, 2016). We explored whether antivirals are reducing LOX enzyme RNA leading to a decrease in ECM stiffness and subsequent delay to ECM detachment. Our results demonstrated no reduction in LOX or LOXL1-4 expression and this lack of a decrease in LOX expression concurs with the AFM data from Fig. 4F, in that if the matrix elastic modulus does not change then we should not see a change in crosslinking enzyme expression with drug treatment. We also studied procollagen N- and C-terminal proteases necessary for collagen processing and fibril formation and therefore incorporation into the matrix. BMP1, a procollagen I C-proteinase and an important regulator of the TGFβ pathway (Ge and Greenspan, 2006; Hopkins et al, 2007) and wound healing (Muir et al, 2016), did not show change at the mRNA level, and neither did ADAMTS2, a procollagen I N-proteinase. We also investigated matrix metalloproteinases (MMPs) since they have been identified as being upregulated in RDEB compared to normal control skin (Gruber et al, 2011; Hata et al, 2015; Kivisaari et al, 2008; Nissinen et al, 2016; Rashidghamat and McGrath, 2017; Riihilä et al, 2021; Stricklin et al, 1982; Titeux et al, 2008) and saw no change upon daclatasvir treatment in 4 separate populations assessed by qPCR. We did see an increase in one population after daclatasvir treatment assessed by RNA-sequencing, suggesting that MMP1 may be a target but with a heterogeneous response in patients. In addition, daclatasvir has been shown previously to downregulate MMP9 in rat models of hepatic fibrosis (Zakaria and El-Sisi, 2020), along with other immunoregulatory factors, but did not have an effect on MMP9 mRNA or protein in the RDEB fibroblasts (Appendix Fig. S5C). Overall, we show that daclatasvir downregulates TGFβ signaling both in vitro and in vivo.

Therapy development for RDEB has been accelerating over the past decade and the first two therapies to be approved focus on treating wounds; a gene therapy approach to deliver type VII collagen (Gurevich et al, 2022) and a bioactive gel to accelerate wound closure (Kern et al, 2023). However, there is need for systemic treatments since RDEB affects internal mucosae, including the esophagus, and wounding can involve a large proportion of body surface area and involve organs such as the eye. Systemic therapies are being explored and clinical trials of losartan, an angiotensin-II receptor antagonist, with published success as an anti-fibrotic in RDEB are ongoing (Nyström et al, 2015; Pourani et al, 2022). While losartan is a promising drug for fibrosis alleviation for RDEB patients, it has been pointed out that losartan cannot be given to pregnant or breast-feeding mothers due to potentially harmful effects to the fetus or newborn, and it is also not recommended for children younger than six years old (Burnier and Wuerzner, 2011; Sica et al, 2005; Yoshinaga, 1999). In addition, preclinical assessment of losartan did not show improvement in survival suggesting that while aspects of fibrosis are mitigated, overall quality of life may not be. There are also preclinical efforts to assess the TGFβ inhibitor decorin, which has demonstrated a reduction in fibrosis and a nonsignificant increase in survival in RDEB mouse models (Cianfarani et al, 2019).

Overall, our study provides the rationale for the use of daclatasvir as a non-invasive, potentially inexpensive (Hill et al, 2016), and reliable therapy option for RDEB that has previously been FDA-approved for treatment in HCV and which shows preclinical efficacy in a mouse model of RDEB.

## Methods

### Cell culture

All patient-derived fibroblasts were collected with informed written consent from each patient in compliance with the Declaration of Helsinki and the experiments conformed to the principles set out in the Department of Health and Human Services Belmont Report. RDEB patient samples are usually donations from biopsy sites, either of non-SCC or SCC origin, and the non-EB patient samples are from breast reduction skin donations from elective surgeries. Cells were isolated from skin

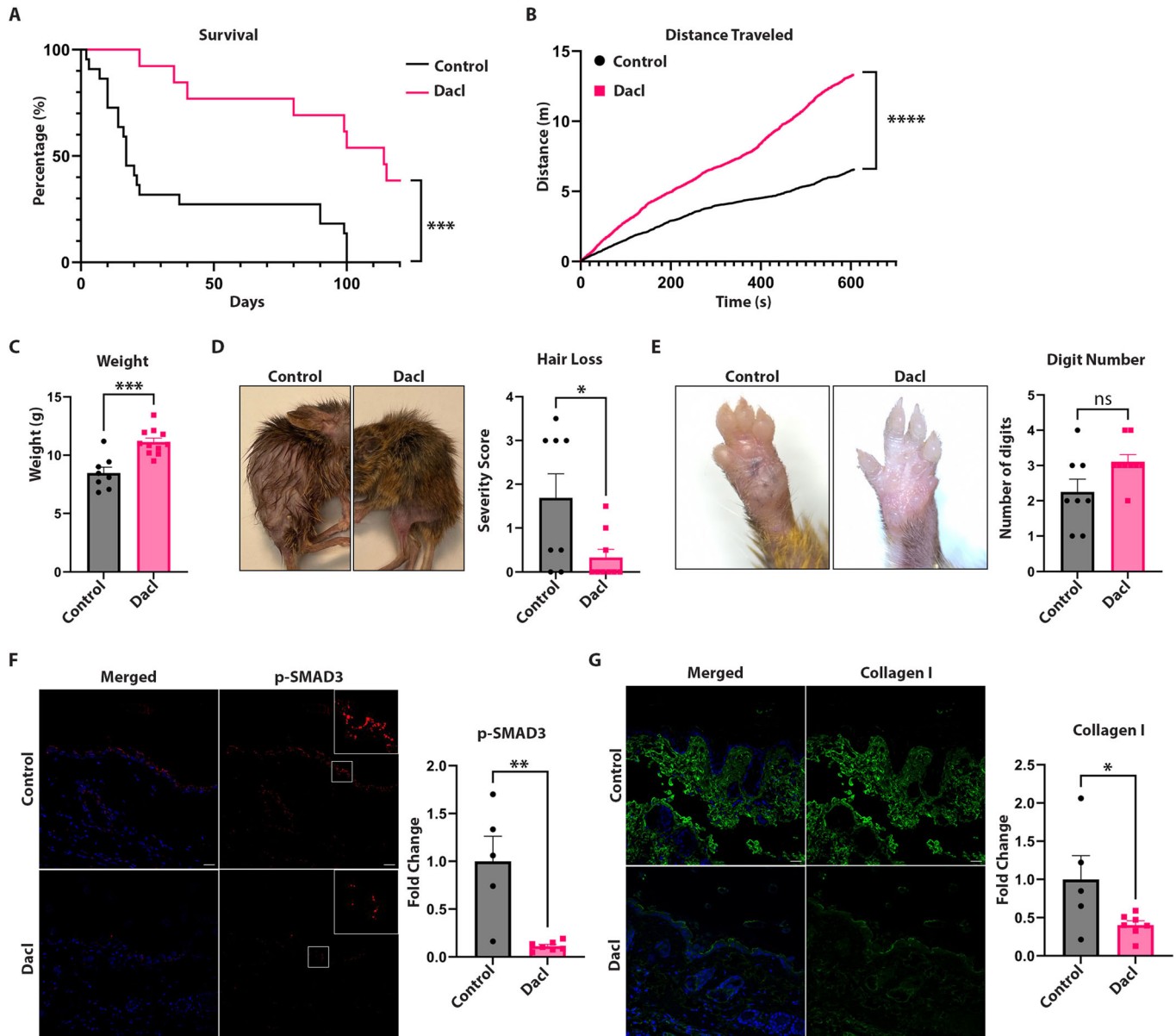

**Figure 5.   Daclatasvir prevents fibrotic progression in RDEB mice and leads to increased longevity and life quality.**

(**A**) Survival curve between daclatasvir-treated RDEB ($n = 13$) mice versus untreated control ($n = 22$) RDEB mice. Kaplan–Meier curve's significance is calculated using the Mantel–Cox test. Mantel–Haenszel hazard ratio (Control/Daclatasvir): 4.910. (**B**) Graph showing distance traveled over 10 min in open-field assay at 60 days old between control ($n = 5$) and daclatasvir-treated ($n = 5$) RDEB mice. Unpaired $t$ test for significance. (**C**) Graph of weights of RDEB mice upon death, comparing untreated control mice ($n = 8$) and daclatasvir-treated RDEB mice ($n = 11$). Unpaired $t$ test for significance. (**D**) Representative images of control ($n = 8$) and daclatasvir-treated ($n = 9$) RDEB mice (left). Score graph shows hair loss severity, unpaired $t$ test for significance (right). (**E**) Representative images of control ($n = 8$) and daclatasvir-treated ($n = 9$) RDEB mice. Graph shows digit loss, unpaired $t$ test for significance (right), $P = 0.0502$. (**F**) Confocal images for immunostaining in dorsal skin of RDEB mice with ($n = 7$) and without ($n = 5$) daclatasvir treatment. Scale bars represent 20 µm in length. Graph of quantification of p-SMAD3 signaling. Unpaired $t$ test for significance. (**G**) Confocal images for immunostaining in the dorsal skin of RDEB mice with ($n = 7$) and without ($n = 5$) daclatasvir treatment. Scale bars represent 20 µm in length. Graph of quantification of collagen I signaling. Unpaired $t$ test for significance. Data information: In (**A–G**), data are presented as mean ± SEM. *$P \leq 0.05$, **$P \leq 0.01$, ***$P \leq 0.001$, ****$P \leq 0.0001$, $P > 0.05$ is not significant (ns). Source data are available online for this figure.

biopsies taken from routine or tumor excision surgeries and cultured at 37 °C at 5% $CO_2$. Primary fibroblasts were grown in Dulbecco's modified essential medium (DMEM, Corning Cellgro, Mediatech Inc, Manassas, VA) with 10% fetal bovine serum (FBS, PEAK Serum, Cat PS-FB1, Colorado, USA) and 1% Penicillin Streptomycin 100X Solution (Corning, Cat 30-002-CI, Manassas,

VA, USA). All media in experimental conditions contained L-ascorbic acid (150 µM, Wako, 012-12061). Non-EB and RDEB fibroblasts (Appendix Table S3) were used up to passage 7. Fibroblasts were plated in 6-well plates to achieve 100% confluence within 2 days of seeding, followed by protein extraction or RNA isolation.

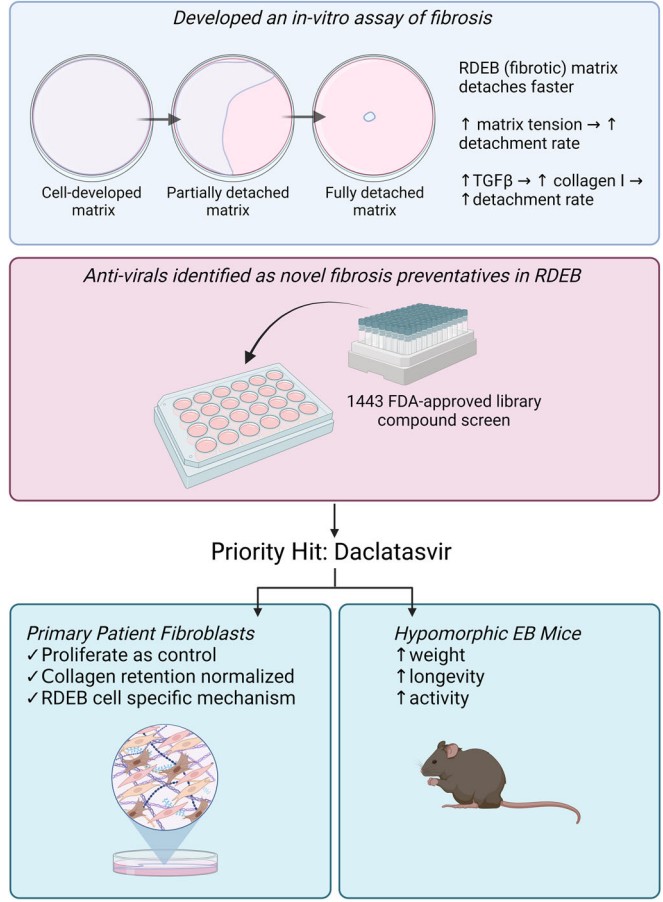

**Figure 6. Daclatasvir demonstrates strong preclinical efficacy in RDEB primary cells and an RDEB mouse model.**

Endogenously-secreted matrix from primary RDEB patient fibroblasts can provide a model for medium-throughput screening of compounds for anti-fibrotic effect. Antivirals were identified as having a novel preventative effect on the dermal fibroblast matrix in RDEB. Daclatasvir, our hit compound of interest, inhibits fibrosis in vitro and in vivo through the modulation of the TGFβ pathway and impacting collagen in the matrix in an RDEB-specific mechanism. The RDEB mouse model experiences improved quality of life and longevity with daclatasvir treatment.

## Detachment assay

Fibroblasts were seeded at 50,000 cells/well in 24-well plates (Corning) and treated with 150 µM L-ascorbic acid in fresh 10% FBS media 24 h after seeding with or without additional treatment. Treated media was changed 3 days per week until the extracellular matrix formed an opaque layer and detached. Media was changed until matrix detached or until 30 days elapsed since ascorbic acid-treatment started. Our screen "hits" were determined by compounds that delayed detachment by more than 2 days after detachment of the DMSO vehicle control.

## Atomic force microscopy

Fibroblasts were seeded at 100,000 cells/well in 12-well plates on plastic coverslips (Thermanox), and 24 h after seeding would be treated with ascorbic acid-treated media for 7 days. Atomic force microscopy (AFM)-based nanoindentation was carried out on fibroblast-derived matrices grown on plastic coverslips in 1× PBS at room temperature using a Dimension Icon AFM (Bruker Nano) and a colloidal spherical tip (radius $R \approx 5$ µm). The spherical tip was prepared by manually

gluing a polystyrene microsphere (PolySciences) onto a tipless silicon nitride cantilever (cantilever C, nominal spring constant k ≈ 0.03 N/m, HQ: CSC38/tipless/Cr-Au, NanoAndMore) with M-Bond 610 epoxy (SPI Supplies) using the same Dimension Icon AFM. At each indentation location, the probe tip was programmed to indent the sample at a 10 µm/s constant z-piezo displacement rate (approximately equals the indentation depth rate) up to a maximum indentation depth of ≈ 1 µm. For each specimen, indentation was performed on relatively flat regions (surface roughness <40 nm for 5 µm × 5 µm contact mode imaging) to minimize the impact of surface roughness. For each sample, at least ten different indentation locations were tested randomly to account for sample spatial heterogeneity. For each indentation curve, the cantilever deflection (in volts) and z-piezo displacement (in µm) were converted to an indentation force (in nN) and depth (in µm) through calibrating the cantilever deflection sensitivity (nm/V) by indenting on a mica substrate and a spring constant (nN/nm) via thermal vibration (Hutter, 1993). The initial tip-sample contact point was determined via an algorithm reported previously for soft materials in the absence of attractive interactions (Han et al, 2011). The loading portion of the resultant force-indentation depth curve was then fit to the finite-thickness corrected Hertz model (Dimitriadis et al, 2002) via

least-squares linear regression to calculate the effective indentation modulus, $E_{ind}$, following our established procedures (Chery et al, 2020; Li et al, 2017), assuming the Poisson's ratio $v \approx 0.5$ (Dimitriadis et al, 2002) and the thickness of fibroblast-derived matrices $\approx 10\ \mu m$.

## MTT assay

We used the MTT Cell Growth Assay Kit (Millipore Sigma, CT02) for all of our viability experiments. Cells were seeded in a 96-well plate at 5000 cells/well and treated for the specified time in each figure legend (up to 10 days). We followed the protocol in the company instructions in the kit and analyzed colorimetric results using the FlexStation 3 plate reader (Molecular Devices).

## Drug treatment

The 1443 FDA-approved drug screening library was purchased from SelleckChem (L1300, FDA-Approved Drug Library), and the Antiviral library was purchased from Tocriscreen (#7350). TGFβ1 recombinant protein was purchased from Cell Signaling (#75362), SB431542 was purchased from Selleckchem (S1067, 10 mM), and SIS3 is from MedChemExpress (HY-13013). Daclatasvir and idoxuridine for in vitro studies were purchased from SelleckChem (S1482 and S1883, respectively). Daclatasvir dihydrochloride for in vivo studies was purchased from MedChemExpress (HY-10465). Cells were treated with 10 µM of each drug from the FDA-approved library or 1 µM of each drug from the Antiviral library (both libraries were diluted in DMSO), 4 µM of TGFβ1, 10 µM of SB431542, 1 µM of SIS3, and 1 µM of daclatasvir and idoxuridine unless otherwise stated. Cells were harvested 48 h after drug treatment.

## Protein quantification

Protein concentration in cell lysate was measured with the Pierce™ Rapid Gold BCA Protein Assay Kit (Thermo Fisher Scientific).

## Immunoblotting

Fibroblasts were seeded at 100,000 cells/well in six-well plates and media containing ascorbic acid media was replaced 24 h later. Fibroblasts were cultured for a further 48 h. Cells were then lysed in RIPA buffer (Thermo Pierce, 89900) supplemented with protease and phosphatase inhibitors (Thermo Pierce tablets, A32963 and A32957, respectively) followed by 3-min centrifugation at 13,300 rpm and 4 °C. Samples were loaded on 8–16% acrylamide gels (Bio-Rad, #4561105). For protein loading, 5 µg of protein was loaded for GAPDH, p-AKT, and collagen I. In total, 25 µg of protein was loaded for p-SMAD3. Primary antibodies include phospho-SMAD3 (Rockland, 600-401-919) at 1:1000 dilution, collagen I (Southern Biotech, 1310-01) at 1:1000 dilution, phosphor-AKT (cell signaling, 4060) at 1:1000 dilution, and GAPDH (Santa Cruz, sc-365062) at 1:2500 dilution.

Proteins were transferred onto a nitrocellulose membrane with a Bio-Rad Trans-Blot-Turbo (Bio-Rad, Hercules, CA), blocked in PBS-0.1% Tween with 5% milk or 5% BSA according to the primary antibody's requirements, and incubated overnight at 4 °C with the primary antibody. After incubation with IgG-HRP conjugated secondary antibody (Cell Signaling), the membrane was incubated with SuperSignal West Femto Maximum Sensitivity Substrate (Fisher Scientific, Waltham, MA) and exposed using the FluorChem R exposure system (ProteinSimple, San Jose, CA). Western blot signals were analyzed using ImageJ. Each protein was quantified relative to GAPDH as the loading control.

## Quantitative polymerase chain reaction (qPCR)

Fibroblasts were seeded at 100,000 cells/well in six-well plates and had their media changed with ascorbic acid-treated media 24 h later. Fibroblasts would have the treated media for 48 h. Total RNA was then isolated using the TRIzol method (Fisher Scientific, Waltham, MA) according to the manufacturer's instructions. RNA extractions were quantified using a NanoDrop One (Fisher Scientific, Waltham, MA) and 1.5 µg RNA was used for cDNA synthesis using SuperScript III First-Strand Synthesis System (Invitrogen, Life Technologies, Carlsbad, CA). Primers designed by from Origene were used, including *GAPDH* (For: GTCTCCTCTGACTTCAACAGCG, Rev: ACCACC CTGTTGCTGTAGCCAA), *COL1A1* (For: GATTCCCTGGACCTA AAGGTGC, Rev: AGCCTCTCCATCTTTGCCAGCA), *TGFB1* (For: TACCTGAACCCGTGTTGCTCTC, Rev: GTTGCTGAGGTATCGC CAGGAA), *LOX* (For: GATACGGCACTGGCTACTTCCA, Rev: GCCAGACAGTTTTCCTCCGCC), *LOXL1* (For: ACAGCACCTGT GACTTCGGCAA, Rev: CGGTTATGTCGATCCACTGGCA), *LOXL2* (For: TGACTGCAAGCACACGGAGGAT, Rev: TCCGA ATGTCCTCCACCTGGAT), *LOXL3* (For: GCACAGTCTGTGACC GCAAGTG, Rev: CTTCACTCAGGTGGATAGCACC), *LOXL4* (For: CCAAAGACTGGACGCGATAGCT, Rev: GGCAGTTTGTGTCCT CCAGACA), *BMP1* (For: CCAATGGCTACTCTGCTCACATG, Rev: AAGCCATCTCGGACCTCCACAT), *ADAMTS2* (For: TACAAGGA CGCCTTCAGCCTCT, Rev: CCACTTTGCAGTGGCTGTTGTC), and *MMP1* (For: ATGAAGCAGCCCAGATGTGGAG, Rev: TGGTC CACATCTGCTCTTGGCA).

For qPCR, SYBR Select Master mix (Life Technologies, Carlsbad, CA) was used, and cDNA samples were diluted 1:10 to serve as a template. We used the QIAgility robot (Qiagen) for pipetting the samples and the Rotor-Gene Q for the cycler (Qiagen). Experiments were performed in duplicate due to highly accurate robot pipetting. qPCR cycles were initial holds of 50 °C for 2 min and 95 °C for 2 min, 35 cycles of annealing (95 °C for 15 s, 60 °C for 60 s), 95 °C for 15 s, 60 °C for 15 s, and 95 °C for 15 s.

## Total collagen (hydroxyproline) assay

Please refer to collagen assay protocol from Cao et al (Cao et al, 2022).

## MMP9 ELISA assay

Fibroblasts were seeded at 100,000 cells/well in six-well plates and media containing ascorbic acid media was replaced 24 h later. Fibroblasts were cultured for a further 7 days. Lysate and media were collected according to the manufacturer's guidelines (abcam, ab246539). We followed the protocol before using the FlexStation 3 (Molecular Devices, San Jose, CA) to measure absorbance.

## Mouse studies

Jefferson's Institutional Animal Care & Use Committee (IACUC) provided approval for these mice experiments (protocol number 02141). We followed the protocol of Nystrom et al (Nyström et al,

2015) working with the RDEB mouse model (Fritsch et al, 2008) that found that creating pure congenic lines of C57BL/6 and 129SV before creating an F1 generation of RDEB mice would increase the survival rate. We utilized services from Charles River for speed congenics (Wakeland et al, 1997) at the N3 and N5 generations by selecting heterozygous mice with the highest background of the desired strain. We also followed the animal caretaking protocol from another publication (Chen et al, 2022) that recommended improved husbandry techniques such as specialized bedding and nutritional gel supplements, which all improved our survival output of the RDEB mice to 40–50%. Control mice received regular water throughout the course of the experiment. Mice chosen for the treatment group were given daclatasvir in utero. Heterozygous dams were given 30 mg/kg of daclatasvir dihydrochloride in their drinking water from conception until RDEB mice were weaned at 28 days old. RDEB mice were euthanized when ethically required based on excessive signs of pain or distress, including severe weight loss, uncontrollable shaking, or lack of mobility over an extended period. Hair loss scoring was performed by one blinded investigator on a scale of 1–5, with 1 being no hair loss at all and 5 constituting near to total hair loss on the torso and legs.

## Immunofluorescence

RDEB mice samples were formalin-fixed for 24 h, paraffin-embedded, and cross-sectioned at 5 μm. Samples were baked overnight and rehydrated, followed by antigen retrieval in citrate buffer. Slides were permeabilized in 0.1% Triton-X100 in PBS for 10 min at room temperature, followed by blocking in 5% bovine serum albumin in PBS with 0.1% Tween-20 for 1 h at room temperature. Cells with primary antibodies were incubated overnight at 4 °C which includes phosphor-SMAD3 (abcam, ab52903) at 1:100 dilution and collagen I (Santa Cruz, sc-59772) at 1:100 dilution.

Secondary antibodies, Alexa Fluor 594 goat anti-rabbit (1:800) (Invitrogen, Eugene, OR) and Alexa Fluor 488 goat anti-mouse (1:250) (Invitrogen, Eugene, OR), were applied for 1 h at room temperature. Coverslips were mounted on the slides with DAPI Fluoromount-G (Southern Biotech, #0100-20) and analyzed by confocal microscopy (Nikon A1R Microscope). Images were quantified using ImageJ's histogram feature in the dermis region of the skin. Regions of interest had consistent area sizes between images.

## Open-field assay

Video images were collected from 60-day-old mice over a 10-min period in an open-field chamber with an open top and camera monitor (Digiscan animal activity monitor, AccuScan Instruments, Inc., Columbus, OH) positioned above. One mouse at a time is placed in the center of the chamber, and no stimulus is provided during the test, which is measuring general activity levels and exploration habits of the mice. Videos are analyzed for movement using an open-source algorithm for tracking rodent movement (Zhang et al, 2020).

## RNA-sequencing

Total RNA was extracted from primary patient fibroblasts (seeded at 100,000 cells/well in a six-well plate and treated for 48 h) using RNA STAT-60 according to manufacturer instructions (Amgen)). Each RNA sample was assessed for degradation by Tapestation 4150 (Agilent) and

the average RNA integrity score for the sample set was 9.8. Sequencing libraries were generated with 100 ng each sample using the Illumina Stranded Total RNA Prep with Ribo-Zero Plus (Illumina, Inc) following manufactory instructions. Final library quality control was carried out by evaluating the fragment size on Tapestation 4150 (Agilent) and the average library size is 338 bp, with ~218 bp insert DNA size. The concentration of each library was determined by Qubit HS dsDNA Quantification Assay Kits (Invitrogen), prior to sequencing. Libraries were normalized to 0.5 nM in Illumina RSB (Resuspension Buffer), then pooled evenly. The pooled libraries were denatured with 0.2 N NaOH following Illumina's "Denature and Dilute Guide" and loaded on NovaSeq 6000 for 100 pM final loading concentration. Cluster generation of the denatured libraries was performed according to the manufacturer's specifications (Illumina, Inc) utilizing the NovaSeq SP 200 flow cell. Libraries were clustered appropriately with a 1% PhiX spike-in. Sequencing-by-synthesis was performed on NovaSeq 6000 with paired-end 101 bp reads and 10 bp index reads resulting in 50 million paired-end reads per sample. Sequence read data were processed and converted to FASTQ format for downstream analysis by Illumina BaseSpace analysis software (v2.0.13). Using Partek Flow (Partek, St. Louis, MO, USA), data filtering steps started with pre-alignment QC. The raw reads corresponding to the Illumina RNA-Seq FASTQ files were mapped with Star aligner against the human genome hg38 using STAR 2.7.8a in Partek Flow software. After alignment, we performed post-alignment QC. Aligned reads were then quantified using Partek E/M method (Annotation model: Ensembl Transcripts release 108). We filtered the data by noise reduction, excluding features where the value ≤ 1 in at least 80% of samples. The data were normalized using the DESeq2 normalization method (median ratio). We used differential statistics (gene list) by the normalized DESeq2 counts.

## Statistical analysis

Significance was determined using GraphPad Prism 9. $P < 0.05$ was considered significant and represented with a *$P < 0.01$ was represented with **$P < 0.001$ was represented with *** and $P < 0.0001$ was represented with ****. All data are specified as normalized or not in the figure legends. Descriptive statistical tests, $N$ number, units, central mean, and SEM dispersion are specified in the figures or figure legends as well. Statistical tests were formed as parametric or non-parametric when appropriate. All tests were performed two-sided, and adjustments to alpha levels were reported as corrections in the figure legends. No randomization procedures were conducted, and all mouse experiments were performed with open-label experimental groups. Only hair loss scoring was performed blinded, all other experiments or analyses were performed without blinding. Biological or technical replicates are stated in the figure legends.

## Graphics

Figure 6 and the Synopsis Image were created with Biorender.com.

## For more information

For more information about epidermolysis bullosa and research funding opportunities, please visit the organizations that helped fund this project, including EB Research Partnership (https://www.ebresearch.org/) and EB Medical Research Foundation (https://www.ebmrf.org/).

## The paper explained

### Problem

The primary objective of this controlled laboratory study was to identify compounds that could be repurposed for fibrosis prevention in the disease recessive dystrophic epidermolysis bullosa (RDEB), which currently has no cure.

### Results

We performed a medium-throughput screen of 1443 FDA-approved compounds using an in vitro model of fibrosis based on ECM produced by primary patient fibroblasts and found 43 drugs that normalized disease phenotype. Our hit of interest demonstrated potent pro-survival and anti-fibrotic therapeutic effects in a preclinical mouse model of RDEB.

### Impact

Daclatasvir is a potential therapeutic option that could improve life quality for patients with the devastating disease RDEB.

## Data availability

All study data are included in the article and/or supporting information with the exception of the RNA-sequencing data, which is available at the NCBI Gene Expression Omnibus (GEO) at https://www.ncbi.nlm.nih.gov/geo/ (Accession Series GSE254347).

## Peer review information

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

## Acknowledgements

Opinions, interpretations, conclusions, and recommendations are those of the authors and are not necessarily endorsed by the Department of Defense or the NIH. We would like to thank all the patients who contributed to this study, Dr. Zhijiu Zhong from the Pathology Shared Resource, Dr. Diane Merry for the use of her Open Field Assay apparatus, Dr. Suhao Han and Dr. Sankar Addya from the SKCC Cancer Genomics Shared Resource and the Microscopy Shared Resource. Sidney Kimmel Cancer Center Shared Resources are supported by the NCI grant 5P30CA056036-17, and sequencing on the NovaSeq 6000 is supported by the NCI grant S10 OD025128. Finally, we would like to thank Dr. Alexander Nystrom and Dr. Vicki Chen for their invaluable input regarding the RDEB mice. Our funding sources include the following: EB Research Partnership (APS); EB Medical Research Foundation (APS). Office of the Assistant Secretary of Defense for Health Affairs and the Defense Health Agency J9, Research and Development Directorate, through the Congressionally Directed Medical Research Program under Award No. W81XWH-18-1-0382 (APS); National Science Foundation, Award No. CMMI-1751898 (LH); National Institute of Arthritis and Musculoskeletal and Skin Diseases' T32 Training Grant No. AR052273 (GT); Drexel University's Research and Engineering for Pediatrics by Interdisciplinary Collaboration Leveraging Education and Partnerships for Pediatric Healthcare from the U.S. Department of Education's Graduate Assistance in Areas of National Need Program Fellowship (NP).

## Author contributions

**Grace Tartaglia**: Investigation; Visualization; Methodology; Writing—original draft; Writing—review and editing. **Ignacia Fuentes**: Conceptualization; Investigation; Methodology; Writing—review and editing. **Neil Patel**: Investigation; Writing—review and editing. **Abigail Varughese**: Investigation; Writing—review and editing. **Lauren E Israel**: Investigation; Writing—review and editing. **Pyung Hun Park**: Investigation; Writing—review and editing. **Michael H Alexander**: Investigation; Writing—review and editing. **Shiv Poojan**: Investigation; Writing—review and editing. **Qingqing Cao**: Investigation; Writing—review and editing. **Brenda Solomon**: Investigation; Writing—review and editing. **Zachary M Padron**: Investigation; Writing—review and editing. **Jonathan A Dyer**: Resources; Writing—review and editing. **Jemima E Mellerio**: Resources; Writing—review and editing. **John A McGrath**: Resources; Writing—review and editing. **Francis Palisson**: Resources; Writing—review and editing. **Julio Salas-Alanis**: Resources; Writing—review and editing. **Lin Han**: Supervision; Writing—review and editing. **Andrew P South**: Conceptualization; Resources; Supervision; Funding acquisition; Methodology; Writing—original draft; Project administration; Writing—review and editing.

## Disclosure and competing interests statement

The authors declare no competing interests.

