## [Peer Review File · EMBO Molecular Medicine]

Antiviral drugs prolong survival in murine recessive dystrophic epidermolysis bullosa

Grace Tartaglia, Ignacia Fuentes, Neil Patel, Abigail Varughese, Lauren Israel, Pyung Hun Park, Michael Alexander, Shiv Poojan, Qingqing Cao, Brenda Solomon, Zachary Padron, Jonathan Dyer, Jemima Mellerio, John McGrath, Francis Palisson, Julio Salas-Alanis, Lin Han, and Andrew South

Corresponding authors: Andrew South (andrew.south@jefferson.edu) , Grace Tartaglia (grace.tartaglia@jefferson.edu)

Review Timeline:

Submission Date:	30th Sep 23
Editorial Decision:	24th Oct 23
Revision Received:	26th Jan 24
Editorial Decision:	9th Feb 24
Revision Received:	16th Feb 24
Accepted:	19th Feb 24

Editor: Zeljko Durdevic

Transaction Report:

24th Oct 2023

Dear Dr. South,

Thank you for the submission of your manuscript to EMBO Molecular Medicine. We have now received feedback from the three reviewers who agreed to evaluate your manuscript. All three referees recognize potential interest of the study but also raise important criticism that should be addressed in a major revision. If you would like to discuss further the points raised by the referees, I am available to do so via email or video. Let me know if you are interested in this option.

We would welcome the submission of a revised version within three months for further consideration. Please let us know if you require longer to complete the revision.

I look forward to receiving your revised manuscript.

Yours sincerely,

Zeljko Durdevic

We require:

- 1) A .docx formatted version of the manuscript text (including legends for main figures, EV figures and tables). Please make sure that the changes are highlighted to be clearly visible.
- 2) Individual production quality figure files as .eps, .tif, .jpg (one file per figure). For guidance, download the 'Figure Guide PDF': (<https://www.embopress.org/page/journal/17574684/authorguide#figureformat>).
- 3) A .docx formatted letter INCLUDING the reviewers' reports and your detailed point-by-point responses to their comments. As part of the EMBO Press transparent editorial process, the point-by-point response is part of the Review Process File (RPF), which will be published alongside your paper.
- 4) A complete author checklist, which you can download from our author guidelines (<https://www.embopress.org/page/journal/17574684/authorguide#submissionofrevisions>). Please insert information in the checklist that is also reflected in the manuscript. The completed author checklist will also be part of the RPF.
- 5) Please note that all corresponding authors are required to supply an ORCID ID for their name upon submission of a revised manuscript.
- 6) It is mandatory to include a 'Data Availability' section after the Materials and Methods. Before submitting your revision, primary datasets produced in this study need to be deposited in an appropriate public database, and the accession numbers and

database listed under 'Data Availability'. Please remember to provide a reviewer password if the datasets are not yet public (see <https://www.embopress.org/page/journal/17574684/authorguide#dataavailability>).

13) Author contributions: You will be asked to provide CRediT (Contributor Role Taxonomy) terms in the submission system. These replace a narrative author contribution section in the manuscript.

14) A Conflict of Interest statement should be provided in the main text.

15) Every published paper now includes a 'Synopsis' to further enhance discoverability. Synopses are displayed on the journal

webpage and are freely accessible to all readers. They include a short stand first (maximum of 300 characters, including space) as well as 2-5 one-sentences bullet points that summarizes the paper. Please write the bullet points to summarize the key NEW findings. They should be designed to be complementary to the abstract - i.e. not repeat the same text. We encourage inclusion of key acronyms and quantitative information (maximum of 30 words / bullet point). Please use the passive voice. Please attach these in a separate file or send them by email, we will incorporate them accordingly.

Please note: When submitting your revision you will be prompted to enter your funding and payment information. This will allow Wiley to send you a quote for the article processing charge (APC) in case of acceptance. This quote takes into account any reduction or fee waivers that you may be eligible for. Authors do not need to pay any fees before their manuscript is accepted and transferred to the publisher.

EMBO Press participates in many Publish and Read agreements that allow authors to publish Open Access with reduced/no publication charges. Check your eligibility: <https://authorservices.wiley.com/author-resources/Journal-Authors/open-access/affiliation-policies-payments/index.html>

***** Reviewer's comments *****

Referee #1 (Remarks for Author):

Dystrophic epidermolysis bullosa (DEB) is a devastating genetic disorder associated with fibrotic scarring. Tartaglia et al. used a novel in vitro model based on endogenously produced extracellular matrix to screen an FDA-approved compound library and identified antivirals as a drug class not yet known for anti-fibrotic action. Their lead drug candidate, daclatasvir, was then evaluated in a mouse model of DEB, where it increased survival of affected animals and mitigated the disease phenotype. Daclatasvir had been approved for clinical use in combination with other antiviral agents as therapy of chronic hepatitis C, but was withdrawn by the sponsor in 2019. However, Tartaglia et al. also screened a focused 240 antiviral drug library and report 57 hits across two DEB dermal fibroblast populations that delayed matrix detachment. Overall, the paper presents a solid set of data, adding something new and relevant to the field of epidermolysis bullosa research. The results support clinical development of antivirals not only for the treatment of DEB but may also pave the way to tackle the burden of other fibrotic diseases.

The following comments need to be addressed to improve the manuscript:

1. What does Supplemental Fig. 2 show? The PDF lacks a legend with relevant information, e.g., which controls were used.
2. The authors describe the effects of numerous compounds on DEB fibroblasts. That disease phenotype serves as control throughout the paper. It would also be interesting to know what happens if selected compounds are applied to wild-type dermal fibroblasts, and the respective data for two compounds are actually reported in Supplemental Fig. 4. Have the authors looked at this more systematically? Any effect that is seen also on wild-type fibroblasts would be more than a rescue effect!
3. Why did the authors investigate only individual mRNAs by qPCR? Results of a complete array would have provided relevant additional information.
4. Clarity for the nonspecialist reader could be improved, especially in the results section.

Referee #2 (Comments on Novelty/Model System for Author):

There is a huge unmet need to treat fibrosis in RDEB.
They included a diverse range of RDEB patient and mouse samples.
They tested a large range of molecules with the assay.

Referee #2 (Remarks for Author):

This study developed assays to assess fibrosis and TGFB in RDEB
Please explain any relation between elevated basic FGF found in RDEB and the TGFB and fibrosis
Do these drugs that reduce fibrosis impact basic FGF as this also stimulates fibrosis?
How was quality of life measured in these mice?

Referee #3 (Remarks for Author):

In the manuscript "Antiviral drugs prolong survival in murine recessive dystrophic epidermolysis bullosa" Tartaglia and colleagues describe a screen of FDA approved drugs to identify compounds to be used in RDEB, a skin fragility disorder characterized by high level of fibrosis. As readout they use an in vitro system i.e., the detachment of cell matrices from culture plates. Anti-viral drugs have a positive effect in vitro, reduce fibrosis characteristics, and two lead compounds are further characterized. Finally, the best scoring compound daclatasvir is tested in a preclinical mouse model and has positive effects on fibrosis, life quality and span.

This is an interesting and well written manuscript that follows a trend observable in the last years: the repurposing of drugs to treat rare diseases for which no causal therapies are available. The findings are interesting and promise a swift translation, addressing an important and unmet clinical need. With respect to the presented data several questions remain which should be addressed prior publication.

Major concerns:

- (1) The screen data should be made publicly available in form of a supplemental excel spreadsheet. How were selections made? Right now, one cannot recapitulate how drugs were selected and which quantitative measures/cutoffs and/or statistics were used to define targets. This is absolutely critical so that readers can perform their own statistical analyses (which may yield different shortlists).
- (2) I cannot follow the logic of hydroxyproline measurements (Figure 1I and 4F). What is the major conclusion of these experiments and how does this relate to literature knowledge? When there is more hydroxyproline in the medium of non-EB cells does it mean that there is more secreted/soluble collagen? This is contradicting the fibrosis phenotype. Authors should perform additional tests e.g., western blots or proteomics to directly address collagen levels in medium. Hydroxyproline is just a proxy and may be present in other proteins or in a free form. It could also indicate a difference in collagen crosslinking and reflect hydroxyproline accessibility (more than collagen amount as misleadingly indicated by the axis labels). Also, axis labels should be corrected and indicate the actual measurement.
- (3) Used drug concentrations in screen: how do the 10⁻⁶ M relate to clinical relevant concentrations? Are drugs prescribed in a similar range, are serum levels in a similar range?
- (4) Figure 4D: data spread of controls should be considered. It appears that control values have been set/normalized to 1. If this is done before significance testing, most changes might wrongly appear significant.

Minor points:

- (1) In Figure 4A, 1⁻⁶ M of daclatasvir seems to have not a significant effect, in contrast to Figure 4B. How are these two figure panels related? How is the difference explained?
- (2) In the discussion a Figure 4H with AFM data is mentioned which is not included in the manuscript. This should be corrected. It would be interesting to compare AFM measurements of EB and non-EB matrices.

Referee #1 (Remarks for Author):

Dystrophic epidermolysis bullosa (DEB) is a devastating genetic disorder associated with fibrotic scarring. Tartaglia et al. used a novel in vitro model based on endogenously produced extracellular matrix to screen an FDA-approved compound library and identified antivirals as a drug class not yet known for anti-fibrotic action. Their lead drug candidate, daclatasvir, was then evaluated in a mouse model of DEB, where it increased survival of affected animals and mitigated the disease phenotype. Daclatasvir had been approved for clinical use in combination with other antiviral agents as therapy of chronic hepatitis C, but was withdrawn by the sponsor in 2019. However, Tartaglia et al. also screened a focused 240 antiviral drug library and report 57 hits across two DEB dermal fibroblast populations that delayed matrix detachment. Overall, the paper presents a solid set of data, adding something new and relevant to the field of epidermolysis bullosa research. The results support clinical development of antivirals not only for the treatment of DEB but may also pave the way to tackle the burden of other fibrotic diseases.

Reply: We thank the Reviewer for their thorough assessment of our manuscript and we are very happy that the Reviewer appreciates the novelty and clinical potential of our findings. We appreciate the Reviewer's recommendations which we have addressed as detailed below. We believe the additions and modifications suggested by the Reviewer have greatly improved our manuscript.

The following comments need to be addressed to improve the manuscript:

1. What does Supplemental Fig. 2 show? The PDF lacks a legend with relevant information, e.g., which controls were used.

Reply: We apologize for the oversight regarding the legend for **Supplemental Figure 2** which we agree was not adequately describing the data presented. We have replaced the legend for **Supplemental Figure 2** to clearly describe the data we are presenting and we thank the Reviewer for flagging this error.

2. The authors describe the effects of numerous compounds on DEB fibroblasts. That disease phenotype serves as control throughout the paper. It would also be interesting to know what happens if selected compounds are applied to wild-type dermal fibroblasts, and the respective data for two compounds are actually reported in Supplemental Fig. 4. Have the authors looked at this more systematically? Any effect that is seen also on wild-type fibroblasts would be more than a rescue effect!

Reply: We agree with the Reviewer that it is important to determine whether antivirals represent an active process in the absence of a disease phenotype. As the Reviewer pointed out we had looked at our lead hit compounds in normal, wild-type, fibroblasts in **Supplemental Figure 4**. We have now added Western blotting data to confirm that our lead antiviral hit compounds do not affect Collagen I, phosphorylated SMAD3 or Akt at the protein level (new **Supplemental Figure 4**).

3. Why did the authors investigate only individual mRNAs by qPCR? Results of a complete array would have provided relevant additional information.

Reply: We thank the reviewer for the suggestion of running a complete array to look at the effects of daclatasvir and idoxuridine on RDEB fibroblasts. In response, and in addition to our focused qPCR approach to assess collagen processing enzymes we have added RNA sequencing data comparing daclatasvir with vehicle control in RDEB fibroblasts. These new data agree with our observations that

daclatasvir reduces collagen I mRNA and has little effect on the collagen processing enzyme genes we originally presented in the old *Supplemental Figure 5*. We have added the RNA sequencing data to the new *Supplemental Figure 5*.

4. Clarity for the nonspecialist reader could be improved, especially in the results section.

We have reviewed the results section and altered some of the text as indicated with a view to improve readability and explanation of our approach for the non-specialist reader. We hope these improvements adequately address this important comment.

Referee #2 (Comments on Novelty/Model System for Author):

*There is a huge unmet need to treat fibrosis in RDEB.
They included a diverse range of RDEB patient and mouse samples.
They tested a large range of molecules with the assay.*

Reply: We thank the Reviewer for their assessment of our manuscript and their appreciation of the range of compounds and breadth of samples used in our study.

Referee #2 (Remarks for Author):

*This study developed assays to assess fibrosis and TGFB in RDEB
Please explain any relation between elevated basic FGF found in RDEB and the TGFB and fibrosis*

Reply: The Reviewer makes a very good point since FGF has been linked with fibrosis in non-RDEB settings and there is a study from the late 90's identifying increased FGF in urine of RDEB patients (Arbiser *et al*, 1998). Previously reported unbiased screens looking at gene expression or protein changes in RDEB fibroblasts compared to non-RDEB fibroblasts have not identified altered levels of FGF (Küttner *et al*, 2014; Küttner *et al*, 2013; Ng *et al*, 2012; Nyström *et al*, 2013) and this is the reason we had not previously looked at levels in our assays. These dermal fibroblast specific data would indicate a separate source for FGF in RDEB patients, perhaps muscle, connective tissues or the kidney itself since these tissues are reported to have relatively high levels of FG2 when reviewing the Protein Atlas (<https://www.proteinatlas.org/ENSG00000138685-FGF2/tissue>). However, the Reviewer makes an excellent suggestion and in response we have performed Western blotting to compare protein levels of FGF in RDEB and non-EB fibroblasts (**Figure R1**). These experiments found no significant change in fibroblast FGF2 protein levels comparing RDEB and non-EB; however, SB treatment significantly reduced FGF2 expression in EB while TGFβ increased FGF2 expression in non-EB, although this was not significant in the three populations of non-EB cells we analyzed. These data are in line with published studies identifying a relationship between TGFβ and FGF (Strand *et al*, 2014) but do not identify differences between RDEB and non-EB.

Do these drugs that reduce fibrosis impact basic FGF as this also stimulates fibrosis?

Reply: We thank the reviewer for their suggestion to look at the impact of antiviral treatment on FGF levels in RDEB fibroblasts. Western blotting experiments found no significant effect on FGF for our two lead hit antiviral compounds as shown (**Figure R2**).

How was quality of life measured in these mice?

Reply: Quality of life for the RDEB mice was measured using activity (more energy and less pain to move around – see **Figure 5B**), hair retention (a sign of diminished pruritus – see **Figure 5D**), and weight gain (increased appetite – see **Figure 5C**).

Figure R1. TGFβ regulation impacts FGF2 expression but shows no difference between RDEB and non-EB. (Left) Western blots of FGF2 and GAPDH in RDEB (n=4) and non-EB (n=3) fibroblasts with SB or TGFβ treatment. (Middle) Quantification of blot presented as graph showing mean ± SEM of FGF2 expression relative to GAPDH. Ordinary one-way ANOVA with Šidák test for significance. **p<0.01; ns, not significant. (Right) Quantification of blot presented as graph showing mean ± SEM of FGF2 expression relative to GAPDH. Unpaired t-test for significance.

Figure R2. Antivirals do not affect FGF2 expression in RDEB fibroblasts. (Left) Western blots of FGF2 and GAPDH in RDEB fibroblasts (n=4) with vehicle control, idoxuridine, or daclatasvir treatment. (Right) Quantification of blot presented as graph showing mean ± SEM of FGF2 expression relative to GAPDH. RN one-way ANOVA for significance.

Referee #3 (Remarks for Author):

In the manuscript "Antiviral drugs prolong survival in murine recessive dystrophic epidermolysis bullosa" Tartaglia and colleagues describe a screen of FDA approved drugs to identify compounds to be used in RDEB, a skin fragility disorder characterized by high level of fibrosis. As readout they use an in vitro system i.e., the detachment of cell matrices from culture plates. Anti-viral drugs have a positive effect in vitro, reduce fibrosis characteristics, and two lead compounds are further characterized. Finally, the best scoring compound daclatasvir is tested in a preclinical mouse model and has positive effects on fibrosis, life quality and span.

This is an interesting and well written manuscript that follows a trend observable in the last years: the repurposing of drugs to treat rare diseases for which no causal therapies are available. The findings are interesting and promise a swift translation, addressing an important and unmet clinical need.

Reply: We thank the Reviewer for their astute and thorough review of our manuscript and we greatly appreciate the questions, comments and suggestions which we have endeavored to address below. We are happy that the Reviewer believes our study to be well written, interesting and have clinical value.

With respect to the presented data several questions remain which should be addressed prior publication. Major concerns:

(1) The screen data should be made publicly available in form of a supplemental excel spreadsheet. How were selections made? Right now, one cannot recapitulate how drugs were selected and which quantitative measures/cutoffs and/or statistics were used to define targets. This is absolutely critical so that readers can perform their own statistical analyses (which may yield different shortlists).

Reply: We thank the reviewer for their interest in our screen data and have provided a supplemental excel table as part of the **SourceData** for **Figure 3**, as required by the Journal. We have also added more details about our quantitative measures in the methods section in terms of how our selections were made. Essentially, the initial screen was performed in triplicate and the average number of days for detachment of the matrix was used to compare vehicle control. Those compounds that delayed detachment longer than 2 days after control were taken forward in the secondary screen. The screens were observational in nature and did not include statistics to define targets. We have added this detail to the manuscript.

(2) I cannot follow the logic of hydroxyproline measurements (Figure 1I and 4F). What is the major conclusion of these experiments and how does this relate to literature knowledge? When there is more hydroxyproline in the medium of non-EB cells does it mean that there is more secreted/soluble collagen? This is contradicting the fibrosis phenotype. Authors should perform additional tests e.g., western blots or proteomics to directly address collagen levels in medium. Hydroxyproline is just a proxy and may be present in other proteins or in a free form. It could also indicate a difference in collagen crosslinking and reflect hydroxyproline accessibility (more than collagen amount as misleadingly indicated by the axis labels). Also, axis labels should be corrected and indicate the actual measurement.

Reply: We appreciate the reviewers' questions regarding the hydroxyproline assay and we now attempt to provide further clarification in the manuscript. Our original major conclusions from the assay experiments were that 1) RDEB matrices retain a greater proportion of their collagen compared with non-EB matrices, in line with fibrosis in RDEB patients and accelerated detachment of RDEB matrices, 2) Stimulating or inhibiting TGF β in non-EB or RDEB matrices (respectively) reverses the matrix retention trends, and 3) while the antivirals promoted the non-EB collagen retention phenotype in RDEB matrices, Idoxuridine

stimulated more collagen retention in non-EB matrices and that contributed to our decision not to pursue Idoxuridine to in vivo trials.

We agree and now acknowledge in the revised manuscript that hydroxyproline is indeed a surrogate for collagen since other proteins are hydroxylated at their proline residues (particularly proteins secreted by dermal fibroblasts (Küttner *et al.*, 2013). We also acknowledge that the measurements we present are not absolute values since the insolubility of collagen presents challenges for absolute values in the matrices and the stability of collagens and hydroxylated proline residues in the media over the seven days of the experiment will likely be influenced by multiple factors. We have added clarity to the manuscript to address these issues and we have altered the graph axis to reflect the actual measurements as requested. Collagen I levels in the matrix are presented throughout the manuscript and the consistent increase as a measure of fibrosis in our study is the level of collagen I in the matrix. Collagen secreted into the media is soluble and not processed into fibrils or other secondary structures and is subject to degradation by cells in culture and is not the focus of our study since we have used cells own ECM (matrix) detachment from tissue culture to identify antiviral drugs. We have previously published that although the level of collagen I is increased in RDEB, the secretion (measured as a ratio of intracellular to extracellular collagen) is unaltered(Cao *et al*, 2022). Other collagens do show changes in their secretion and intracellular retention, such as collagen 12, but how this relates to matrix composition and detachment is unknown. We feel that a proteomics study of secreted vs ECM retained collagen is beyond the scope of our study and we refer the Reviewer to the excellent work of Jorn Dengjel, Victoria Kuttner and colleagues who report on beautiful proteomics in RDEB and normal fibroblasts (Küttner *et al.*, 2014; Tölle & Dengjel, 2019).

(3) Used drug concentrations in screen: how do the 10 µM relate to clinical relevant concentrations? Are drugs prescribed in a similar range, are serum levels in a similar range?

We appreciate the Reviewer's point and agree that 10µM cannot be considered a clinically relevant concentration for all compounds since this will vary from compound to compound. For the purpose of a screen we started with 10µM and moved forward with our lead hits at 1µM. Our rationale for starting at a relatively high concentration was to avoid taking compounds that could potentially be toxic in the setting of RDEB since patients with RDEB suffer from failure to thrive and often are at weights associated with children, which would lead to increased drug exposure if taking a standard dose of an approved therapy. We wanted to make sure that we didn't pursue drugs with potential side effects in an already challenging clinical setting. For the animal work we do use a physiologically and clinical relevant dose in line with the dose used clinically for daclatasvir (Zakaria & El-Sisi, 2020).

(4) Figure 4D: data spread of controls should be considered. It appears that control values have been set/normalized to 1. If this is done before significance testing, most changes might wrongly appear significant.

Reply: We appreciate the Reviewer's comment that indeed normalizing controls to 1 has the potential to influence significance for certain datasets. In this instance we are analyzing the change in protein abundance comparing treatment with vehicle control using multiple patient primary populations of fibroblasts with Western blotting. Given that there is considerable variation in both baseline signal from separate populations as well as intensities of chemiluminescence, which itself does not always provide consistent values comparing different experiments, the appropriate way to measure change in protein abundance after treatment is to normalize the data to vehicle control. We consulted our local biostatistician, Dr. Tingting Zhan who confirmed that this approach is the correct practice for this type of

analysis. Our raw data is available for review in the **SourceData** associated with each figure as per journal policy.

Minor points:

(1) In Figure 4A, 1 μ M of daclatasvir seems to have not a significant effect, in contrast to Figure 4B. How are these two figure panels related? How is the difference explained?

Reply: We thank the reviewer for pointing out this discrepancy which was a result of completely separate experiments performed at very different times. The original experiments shown in the **old Figure 4A** were used to determine the best dose moving forward from the screen. 1 μ M was identified and taken forward, and the **old Figure 4B** confirmed that this dose indeed consistently delayed detachment in 6 separate RDEB populations, comparing to vehicle alone control each time. Whilst daclatasvir consistently delayed detachment in the **old Figure 4A** the data were under powered to reach significance. As a result, we have moved the **old Figure 4A** to the **new Supplementary Figure 3**. The **new Figure 4** now begins with the original separate, and suitably powered experiments using 6 different RDEB populations showing that our lead screen hits delay detachment in our in vitro assay of fibrosis.

(2) In the discussion a Figure 4H with AFM data is mentioned which is not included in the manuscript. This should be corrected. It would be interesting to compare AFM measurements of EB and non-EB matrices.

Reply: We thank the reviewer for catching the typo and have corrected the discussion to point out **Figure 4G (new 4F)**. We have also shown AFM measurements of RDEB and non-EB matrices in **Figure 1F**.

References

- Arbiser JL, Fine JD, Murrell D, Paller A, Connors S, Keough K, Marsh E, Folkman J (1998) Basic fibroblast growth factor: a missing link between collagen VII, increased collagenase, and squamous cell carcinoma in recessive dystrophic epidermolysis bullosa. *Mol Med* 4: 191-195
- Cao Q, Tartaglia G, Alexander M, Park PH, Poojan S, Farshchian M, Fuentes I, Chen M, McGrath JA, Palisson F *et al* (2022) Collagen VII maintains proteostasis in dermal fibroblasts by scaffolding TANGO1 cargo. *Matrix Biol* 111: 226-244
- Küttner V, Mack C, Gretzmeier C, Bruckner-Tuderman L, Dengjel J (2014) Loss of collagen VII is associated with reduced transglutaminase 2 abundance and activity. *J Invest Dermatol* 134: 2381-2389
- Küttner V, Mack C, Rigbolt KT, Kern JS, Schilling O, Busch H, Bruckner-Tuderman L, Dengjel J (2013) Global remodelling of cellular microenvironment due to loss of collagen VII. *Mol Syst Biol* 9: 657
- Ng YZ, Pourreyron C, Salas-Alanis JC, Dayal JH, Cepeda-Valdes R, Yan W, Wright S, Chen M, Fine JD, Hogg FJ *et al* (2012) Fibroblast-derived dermal matrix drives development of aggressive cutaneous squamous cell carcinoma in patients with recessive dystrophic epidermolysis bullosa. *Cancer Res* 72: 3522-3534
- Nyström A, Velati D, Mittapalli VR, Fritsch A, Kern JS, Bruckner-Tuderman L (2013) Collagen VII plays a dual role in wound healing. *J Clin Invest* 123: 3498-3509

Strand DW, Liang YY, Yang F, Barron DA, Ressler SJ, Schauer IG, Feng XH, Rowley DR (2014) TGF- β induction of FGF-2 expression in stromal cells requires integrated smad3 and MAPK pathways. *Am J Clin Exp Urol* 2: 239-248

Tölle RC, Dengjel J (2019) Effects of the Extracellular Matrix on the Proteome of Primary Skin Fibroblasts. *Methods Mol Biol* 1993: 193-204

Zakaria S, El-Sisi AE (2020) Daclatasvir and sofosbuvir mitigate hepatic fibrosis through downregulation of TNF- α / NF- κ B signaling pathway. *Curr Mol Pharmacol*

9th Feb 2024

Dear Dr. South,

Thank you for the submission of your revised manuscript to EMBO Molecular Medicine. I am pleased to inform you that we will be able to accept your manuscript pending the following final amendments:

1) Figures:

- Remove all figures from the main manuscript file and leave only their legends placed after "References". All supplementary figures should be moved to Appendix file and renamed to Appendix Figure S1 etc. For more information on figure presentation please check "Author Guidelines". <https://www.embopress.org/page/journal/17574684/authorguide#datapresentationformat>
- We notice that Western blot images are of low resolution. Please display images in all figures in higher resolution.
- Figures R1 and R2 are meant for referees, is that correct? If this is the case and since these figures are also shown in PbP, please remove them in the next submission.

2) In the main manuscript file, please do the following:

- Please address all comments suggested by our data editors listed below:

o Figure legends:

1. Please note that a separate 'Data Information' section is required in the legends of figures 1a-h; 2b-f; 3d-g, k; 4a, c-d, f; 5a-g.
2. Please define the annotated p values ** in the legend of supplementary figure 4c, as appropriate.
3. Please indicate the statistical test used for data analysis in the legends of figures 5f-g.
4. Please note that in figures 1a-d, f-g; 2b-f; 4a-b, c, f; 5a-d, f-g; there is a mismatch between the annotated p values in the figure legend and the annotated p values in the figure file that should be corrected.
5. Please note that information related to n is missing in the legends of figures 1i; 3d-k, supplementary figure 2b.
6. Please note that n=2 in supplementary figure 3.
7. Although 'n' is provided, please describe the nature of entity for 'n' in the legends of figures 1a-i; 2f; 4a-f, supplementary figures 3; 4a-c; 5a, c.
8. Please note that the error bars are not defined in the legends of figures 3d-k, supplementary figure 2b.

- Limit keywords to max. 5.

- Add callouts for Figures 2F and 5G.

- Remove abbreviation list.

- In M&M, provide the statement that in addition to the WMA Declaration of Helsinki the experiments also conformed to the principles set out in the Department of Health and Human Services Belmont Report.

- In M&M, statistical paragraph should reflect all information that you have filled in the Authors Checklist, especially regarding randomization, blinding, replication.

- Please rename "Conflicts of interest" to "Disclosure Statement & Competing Interests". We updated our journal's competing interests policy in January 2022 and request authors to consider both actual and perceived competing interests. Please review the policy <https://www.embopress.org/competing-interests> and update your competing interests if necessary.

- Author contributions: Please remove it from the manuscript and specify author contributions in our submission system. CRediT has replaced the traditional author contributions section because it offers a systematic machine-readable author contributions format that allows for more effective research assessment. You are encouraged to use the free text boxes beneath each contributing author's name to add specific details on the author's contribution. More information is available in our guide to authors:

<https://www.embopress.org/page/journal/17574684/authorguide#authorshipguidelines>

3) Data availability: Please make sure that all data deposited in public repositories are freely accessible upon publication.

4) Appendix: Please move all supplementary figures and tables with their legends to Appendix and upload it as a single pdf file with a table of content and page numbers on the first page. Rename supplementary figures and tables to Appendix Figure S1 etc. and Appendix Table S1 etc, also in the main manuscript text.

5) Funding: Please merge it with the "Acknowledgements".

6) The Paper Explained: Disease name should be written out in "Problem" section.

Problem: The primary objective of this controlled laboratory study was to identify compounds that could be repurposed for fibrosis prevention in recessive dystrophic epidermolysis bullosa (RDEB), which currently has no cure.

7) Synopsis: Every published paper now includes a "Synopsis" to further enhance discoverability. Synopses are displayed on the journal webpage and are freely accessible to all readers. They include separate synopsis image and synopsis text.

- Synopsis image: Please provide a striking image or visual abstract as a high-resolution jpeg file 550 px-wide x (250-400)-px high to illustrate your article.

- Synopsis text: Please provide a short standfirst (maximum of 300 characters, including space) as well as 2-5 one sentence bullet points that summarise the paper as a .doc file. Please write the bullet points to summarise the key NEW findings. They should be designed to be complementary to the abstract - i.e. not repeat the same text. We encourage inclusion of key acronyms and quantitative information (maximum of 30 words / bullet point). Please use the passive voice.

- Please check your synopsis text and image before submission with your revised manuscript. Please be aware that in the proof

stage minor corrections only are allowed (e.g., typos).

8) Source data: Please upload one file per figure.

9) For more information: This space should be used to list relevant web links for further consultation by our readers. Could you identify some relevant ones and provide such information as well? Some examples are patient associations, relevant databases, OMIM/proteins/genes links, author's websites, etc...

10) As part of the EMBO Publications transparent editorial process initiative (see our Editorial at <http://embomolmed.embopress.org/content/2/9/329>), EMBO Molecular Medicine will publish online a Review Process File (RPF) to accompany accepted manuscripts. This file will be published in conjunction with your paper and will include the anonymous referee reports, your point-by-point response and all pertinent correspondence relating to the manuscript. Let us know whether you agree with the publication of the RPF and as here, if you want to remove or not any figures from it prior to publication. Please note that the Authors checklist will be published at the end of the RPF.

11) Please provide a point-by-point letter INCLUDING my comments as well as the reviewer's reports and your detailed responses (as Word file).

I look forward to reading a new revised version of your manuscript as soon as possible.

Yours sincerely,

Zeljko Durdevic

*** Instructions to submit your revised manuscript ***

1) a .docx formatted version of the manuscript text (including Figure legends and tables)

2) Separate figure files*

3) supplemental information as Expanded View and/or Appendix. Please carefully check the authors guidelines for formatting Expanded view and Appendix figures and tables at <https://www.embopress.org/page/journal/17574684/authorguide#expandedview>

4) a letter INCLUDING the reviewer's reports and your detailed responses to their comments (as Word file).

5) The paper explained: EMBO Molecular Medicine articles are accompanied by a summary of the articles to emphasize the major findings in the paper and their medical implications for the non-specialist reader. Please provide a draft summary of your article highlighting

This may be edited to ensure that readers understand the significance and context of the research.

Please refer to any of our published articles for an example.

6) For more information: There is space at the end of each article to list relevant web links for further consultation by our readers. Could you identify some relevant ones and provide such information as well? Some examples are patient associations, relevant databases, OMIM/proteins/genes links, author's websites, etc...

7) Author contributions: the contribution of every author must be detailed in a separate section.

8) EMBO Molecular Medicine now requires a complete author checklist (<https://www.embopress.org/page/journal/17574684/authorguide>) to be submitted with all revised manuscripts. Please use the checklist as guideline for the sort of information we need WITHIN the manuscript. The checklist should only be filled with page numbers where the information can be found. This is particularly important for animal reporting, antibody dilutions (missing) and exact values and n that should be indicated instead of a range.

9) Every published paper now includes a 'Synopsis' to further enhance discoverability. Synopses are displayed on the journal webpage and are freely accessible to all readers. They include a short stand first (maximum of 300 characters, including space) as well as 2-5 one sentence bullet points that summarise the paper. Please write the bullet points to summarise the key NEW findings. They should be designed to be complementary to the abstract - i.e. not repeat the same text. We encourage inclusion of key acronyms and quantitative information (maximum of 30 words / bullet point). Please use the passive voice. Please attach these in a separate file or send them by email, we will incorporate them accordingly.

You are also welcome to suggest a striking image or visual abstract to illustrate your article. If you do please provide a jpeg file 550 px-wide x 300-800px high.

10) A Conflict of Interest statement should be provided in the main text

11) Please note that we now mandate that all corresponding authors list an ORCID digital identifier. This takes <90 seconds to complete. We encourage all authors to supply an ORCID identifier, which will be linked to their name for unambiguous name identification.

Currently, our records indicate that the ORCID for your account is 0000-0001-7650-0835.

Link Not Available

Photos 400-800 DPI

*Additional important information regarding figures and illustrations can be found at

<https://bit.ly/EMBOPressFigurePreparationGuideline>. See also figure legend preparation guidelines:

<https://www.embopress.org/page/journal/17574684/authorguide#figureformat>

***** Reviewer's comments *****

Referee #1 (Remarks for Author):

The authors have addressed my comments appropriately and consistently. The manuscript is now suitable for publication.

Referee #3 (Remarks for Author):

The authors addressed all my concerns. I congratulate them to an interesting paper and hope for swift translation of the findings.

The authors addressed the remaining editorial issues.

19th Feb 2024

Dear Dr. South,

We are pleased to inform you that your manuscript is accepted for publication and is now being sent to our publisher to be included in the next available issue of EMBO Molecular Medicine.
